# MAX-DOAS observations of formaldehyde and nitrogen dioxide at three sites in Asia and comparison with the global chemistry transport model CHASER

Hossain Mohamed Syedul Hoque[1], Kengo Sudo[1,2], Hitoshi Irie[3], Alessandro Damiani[3], Manish Naja[4], and Al Mashroor Fatmi[3]

[1]Graduate School of Environmental Studies, Nagoya University, Nagoya, 4640064, Japan

[2]Japan Agency for Marine-Earth Science and Technology (JAMSTEC), Kanagawa, 2370061, Japan

[3]Center for Environmental Remote Sensing (CEReS), Chiba University, Chiba,2638522, Japan

[4]Aryabhatta Research Institute for Observational Sciences (ARIES), Manora Peak, Nainital-263001, Uttarakhand, India

*Correspondence to*: Hossain Mohammed Syedul Hoque (hoque.hossain.mohammed.syedul.u6@f.mail.nagoya-u.ac.jp or hoquesyedul@gmail.com)

**Abstract.** Formaldehyde (HCHO) and nitrogen dioxide ($NO_2$) concentrations and profiles were retrieved from ground-based multi-axis differential optical absorption spectroscopy (MAX-DOAS) observations during January 2017 - December 2018 at three sites in Asia: (1) Phimai (15.18°N, 102.5°E), Thailand; (2) Pantnagar (29°N, 78.90°E) in the Indo Gangetic Plain (IGP), India; and (3) Chiba (35.62°N, 140.10°E), Japan. Retrievals were performed using the Japanese MAX-DOAS profile retrieval algorithm ver. 2 (JM2). The observations were used to evaluate the $NO_2$ and HCHO partial columns and profiles (0 - 4 km) simulated using the global chemistry transport model (CTM) CHASER. The $NO_2$ and HCHO concentrations at all three sites showed consistent seasonal variation throughout the investigated period. Biomass burning affected the HCHO and $NO_2$ variations at Phimai during the dry season and at Pantnagar during spring (March - May) and post-monsoon (September - November). Results found for the HCHO to $NO_2$ ratio ($R_{FN}$), an indicator of high ozone sensitivity, indicate that the transition region (i.e., $1 < R_{FN} < 2$) changes regionally, echoing the recent finding for $R_{FN}$ effectiveness. Moreover, reasonable estimates of transition regions can be derived, accounting for the $NO_2$ - HCHO chemical feedback.

The model was evaluated against global $NO_2$ and HCHO columns data retrieved from Ozone Monitoring
Instrument (OMI) observations before comparison with ground-based datasets. Despite underestimation,
the model well simulated the satellite-observed global spatial distribution of $NO_2$ and HCHO, with
respective spatial correlations (*r*) of 0.73 and 0.74. CHASER demonstrated good performance,
reproducing the MAX-DOAS retrieved HCHO and $NO_2$ abundances at Phimai, mainly above 500 m from
the surface. Model results agree with the measured variations within the one sigma standard deviation of
the observations. Simulations at higher resolution improved the modeled $NO_2$ estimates for Chiba,
reducing the mean bias error (MBE) for the 0 - 2 km height by 35%, but resolution-based improvements
were limited to surface layers. Sensitivity studies show that at Phimai, pyrogenic emissions contribute to
HCHO and $NO_2$ concentrations up to 50 and 35%, respectively.

# 1 Introduction

Formaldehyde (HCHO), the most abundant carbonyl compound in the atmosphere, is a high-yield product
of oxidization of all primary volatile organic compounds (VOCs) emitted from natural and anthropogenic
sources by hydroxyl radicals (OH). Oxidation of long-lived VOCs such as methane produces a global
HCHO background concentration of 0.2 – 1.0 ppbv in remote marine environments (Weller et al., 2000;
Burkert et al., 2001; Singh et al., 2004; Sinreich et al., 2005). Aside from oxidation of VOCs, the
significant sources of HCHO are direct emissions from biomass burning, industrial processes, fossil fuel
combustion (Lee et al., 1997; Hak et al., 2005; Fu et al., 2008), and vegetation (Seco et al., 2007).
However, oxidization of non-methane VOCs emitted from biogenic (e.g., isoprene) or anthropogenic (e.g.,
butene) sources govern the spatial variation of HCHO on a global scale (Franco et al., 2015). The sinks
of HCHO include photolysis at wavelengths shorter than 400 nm, oxidation by OH, and wet deposition,
thereby limiting the lifetime of HCHO to a few hours (Arlander et al., 1995).
Nitrogen dioxide ($NO_2$), an important atmospheric constituent, (1) participates in the catalytic
formation of tropospheric ozone ($O_3$), (2) acts as a catalyst for stratospheric ozone ($O_3$) destruction
(Crutzen, 1970), (3) contributes to the formation of aerosols (Jang and Kamens, 2001), (4) acts as a
precursor of acid rain (Seinfeld and Pandis, 1998), and (5) strongly affects radiative forcing (Solomon et
al. 1999; Lelieved et al., 2002;). Nitrogen oxides ($NO_x$ = NO (nitric oxide) + $NO_2$) are emitted from
natural and anthropogenic sources. Primary $NO_x$ emission sources include biomass burning, fossil fuel

combustion, soil emissions, and lightning (Bond et al., 2001; Zhang et al., 2003). Not only do $NO_x$ emissions degrade air quality; they are leading air pollutant (Ma et al., 2013). Both HCHO and $NO_2$ are important intermediates in the global VOC–$HO_x$ (hydrogen oxides)–$NO_x$ catalytic cycle, which governs $O_3$ chemistry in the troposphere (Lee et al., 1997; Houweling et al., 1998; Hak et al., 2005; Kanakidou et al., 2005). Thus, both trace gases play crucially important roles in tropospheric chemistry.

The observational sites examined for the present study have different atmospheric characteristics. Thailand is strongly affected by pollution because of rapid economic development and urbanization. Moreover, biomass burning in Southeast Asia is a significant source of $O_3$ precursors, contributing up to 30% of the total concentrations during the peak burning season (Amnuaylorajen et al., 2020; Khodmanee et al. 2021). Because of rapid industrialization, India the second most populous country in the world, is witnessing an increasing $O_3$ trend along with $NO_2$ and HCHO concentrations in all major cities (Mahajan et al; 2015; Lu et al, 2018;). The Indo-Gangetic Plain (IGP), which covers ~21% of the Indian subcontinent land area is hotspots of severe air pollution (Giles et al; 2005, Biswas et al; 2019). In contrast, surface $O_3$ concentrations have shown an increasing trend in Japan, despite decreasing $NO_x$ and VOC concentrations related to emission control measures after 2000 (Irie et al., 2021). Therefore, observational and modeling studies must be conducted to improve our quantitative understanding of the $O_3$-$NO_x$-VOC relation in these regions.

Multi-axis differential optical absorption spectroscopy (MAX-DOAS), a well-established, unique, and powerful remote sensing method for measuring trace gases and aerosols, is based on the DOAS technique. Aerosols and trace gases are quantified using selective narrowband (high frequency) absorption features (Platt 1994; Platt and Stutz 2008). Spectral radiance measurements at different elevation angles (ELs) can provide profile information about atmospheric trace gases and aerosols (Hönninger et al., 2004; Wagner et al., 2004; Wittrock et al., 2004; Frieß et al., 2006; Irie et al., 2008a). Many studies have demonstrated the retrieval of aerosol and trace gas concentrations and profiles from MAX-DOAS observations, including $NO_2$ and HCHO (Clémer et al., 2010; Irie et al., 2011; Hendrick et al., 2014; Wang et al., 2014; Franco et al., 2015; Frieß et al., 2016).

The ability of MAX-DOAS to provide information related to surface concentrations, vertical profiles, and column densities makes it a good complement to ground-based in situ and satellite observations.

Moreover, the MAX-DOAS method uses narrowband absorption of the target compounds, thereby obviating any need for radiometric calibration of the instrument. Because of these advantages, MAX-DOAS systems are deployed for the assessment of aerosol and trace gases in regional and global observational networks such as BREDOM (Wittrock et al., 2004), BIRA-IASB (Clémer et al., 2010), and MADRAS (Kanaya et al., 2014). Such datasets are used, in but are not limited to, (1) air quality assessment and monitoring, (2) evaluation of chemistry-transport models (CTMS), and (3) validation of satellite data retrievals. Several studies have used MAX-DOAS datasets to validate tropospheric columns retrieved from satellite observations, including $NO_2$ and HCHO (Irie et al., 2008b; Ma et al., 2013; Chan et al., 2020; Ryan et al., 2020). However, limited MAX-DOAS datasets have been used to evaluate global CTMs. Vigouroux et al. (2009) and Franco et al. (2015) respectively used the MAX-DOAS HCHO datasets from Reunion Island and Jungfraujoch stations to evaluate the Intermediate Model of Annual and Global Evolution of Species (IMAGES) and GEOS-Chem model simulations. Kanaya et al. (2014) validated the Model for Interdisciplinary Research on Climate–Earth System Model – Chemistry (MIROC-ESM-CHEM) simulated $NO_2$ column densities with MAX-DOAS observations in Cape Hedo and Fukue in Japan. Kumar et al. (2021) used MAX-DOAS observations to evaluate the high-resolution regional model Meco(n)(MESSy-field ECHAM and COSMO model nested n times).

For this study, $NO_2$ and HCHO profiles retrieved from MAX-DOAS observations from the International air quality and sky research remote sensing (A-SKY) (http://atmos3.cr.chiba-u.jp/a-sky/) network sites are used to evaluate the global Chemical Atmospheric General Circulation Model for the Study of Atmospheric Environment and Radiative Forcing (CHASER; Sudo et al., 2002). The three A-SKY sites of - (1) Phimai in Thailand (15.18°N, 102.56°E), (2) Pantnagar (29°N, 78.90°E) in the IGP in India, and (3) Chiba (35.62°N, 140.10°E) in Japan, are respectively representative of rural, semi-rural, and urban environments. CHASER has been used mostly for global-scale research (Sudo et al., 2007; Sekiya et al., 2014, 2018; Miyazaki et al., 2017). The study described herein is the first reported attempt to evaluate the CHASER-simulated $NO_2$ and HCHO profiles using MAX-DOAS observations in three different atmospheric environments. Moreover, few reports of the literature have described the use of MAX-DOAS datasets to evaluate global CTMs in South Asia and South-east Asia. Overall, this study was conducted

to provide important insights into model performances and to help reduce model uncertainties related to NO$_2$ and HCHO simulations in these regions.

The paper is structured in the following manner. First, the observation sites, MAX-DOAS instrumentation, and retrieval strategies are described in section 2. Section 2 also includes a short description of the CHASER model and Ozone Monitoring Instrument (OMI) HCHO and NO$_2$ retrievals. Next, the observations and the evaluation results are described in sections 3. Finally, the sensitivity study results are provided in section 3.4. and the concluding remarks in section 4.

## 2 Observations, datasets, and methods

### 2.1 Site Information

Continuous MAX-DOAS observations at Chiba, Phimai, and Pantnagar started respectively in 2012, 2014, and 2017. The measurements from January 2017 to December 2018 at all three sites are discussed herein. Phimai, a rural site, is located ~260 km north-east of the Bangkok metropolitan area and is unlikely to be affected by vehicular and industrial emissions. However, the site is affected by biomass burning during January - April. Two major air streams: the dry, cool north-east monsoon during November – mid-February and the wet, warm south-west monsoon during mid-May – September affect the climate in Phimai. As described by, Hoque et al. (2018), the climate classifications of Phimai are the (a) dry season (January – April), and (b) wet season (June – September).

Pantnagar, a semi-urban site in India, is located in the IGP. The Indian capital of New Delhi is situated at ~225 km south-west of the site. The low-altitude plains are on the south and west sides of the site. The Himalayan mountains are located to the north and east. An important roadway with moderate traffic volume and a small local airport lies within 3 km of the site. Rudrapur (~12 km south-west of Pantnagar) and Haldwani (~ 25 km north-east of Pantnagar) are the two major cities near Pantnagar, where non-combustible industries are located (Joshi et al., 2016). The climate classification at Pantnagar is the

following: (1) winter (December–February), (2) spring (March-May), (3) summer monsoon (June–
August), and (4) autumn (September–November).

Chiba, an urban site, is located ~40 km south-east of the Tokyo metropolitan region. Tokyo Bay,

large-scale industries, and residential areas are located within a 50 km radius. Chiba has four distinct
seasons: (1) spring (March-May), (2) summer (June–August), (3) autumn (September–November), and
winter (December– February). The locations of the three sites are depicted in Fig. 1.


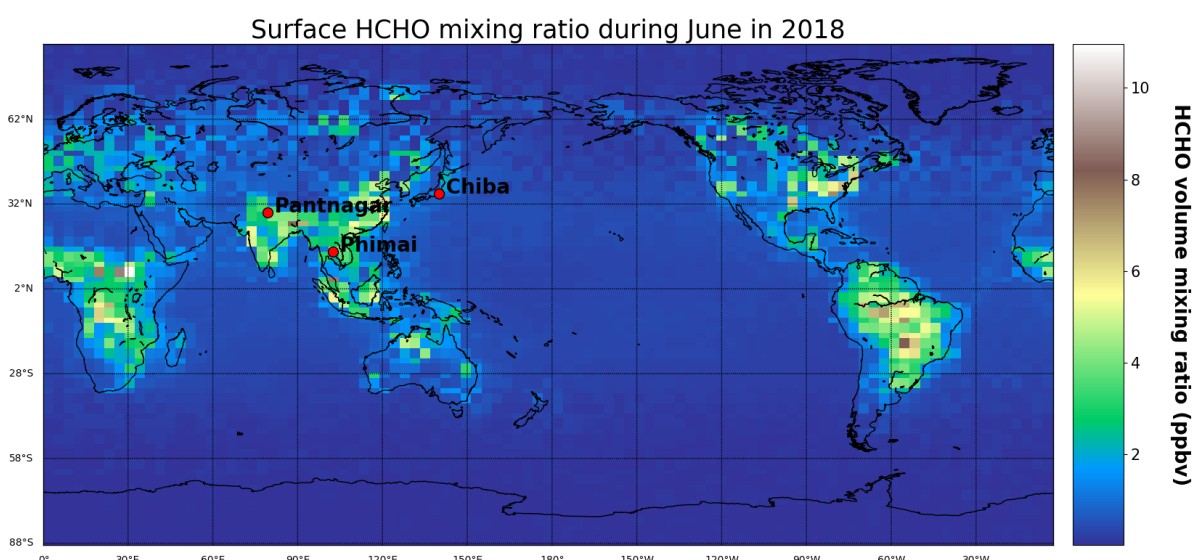

**Figure 1:** Surface HCHO mixing ratio in June 2018, simulated using the CHASER model. The red points
represent the locations of the observation sites, which are part of the A-SKY network.


## 2.2 MAX-DOAS retrieval

The MAX-DOAS systems used for continuous observations at the three sites participated in the Cabauw Intercomparison Campaign of Nitrogen Dioxide measuring Instruments (CINDI) (Roscoe et al., 2010) and CINDI-2 (Kreher et al., 2020) campaigns. The instrumentation setup is described by Irie et al. (2008, 2011, 2015). The indoor part of the MAX-DOAS systems consists of an ultraviolet-visible (UV-VIS) spectrometer (Maya2000Pro; Ocean Optics Inc.) embedded in a temperature-controlled box. The outdoor unit consists of a single telescope and a 45° inclined movable mirror on a rotary actuator, used to perform reference and off-axis measurements. The high-resolution spectra from 310–515 nm is recorded at six elevation angles (ELs) of 2°, 3°, 4°, 6°, 8°, and 70° at the Chiba and Phimai sites. At the Pantnagar site, measurements are conducted at ELs of 3°, 4°, 5°, 6°, 8°, and 70°. The sequences of the ELs at all the sites were repeated every 15 min. The reference spectra are recorded at EL of 70° instead of 90° to avoid saturation of intensity. Because all the ELs were considered in the box air mass factor ($A_{box}$) calculation to retrieve the vertical profile, the choice of reference EL (70˚ or 90˚) is not an important issue for this study. The off-axis ELs are limited to $< 10°$ to reduce the systematic error in the in-oxygen collision complex ($O_4$) fitting results (Irie et al., 2015), thereby maintaining high sensitivity in the lowest layer of the retrieved aerosol and trace gas profiles. Daily wavelength calibration using the high-resolution solar spectrum from Kurucz et al. (1984) is performed to account for the spectrometer's long-term degradation. The spectral resolution (full width half maximum: FWHM) is about 0.4 nm at 357 and 476 nm. The concentrations and profiles of aerosol and trace gases are retrieved using the Japanese vertical profile retrieval algorithm (JM2 ver. 2) (Irie et al., 2011, 2015). The algorithm works in three steps: (1) DOAS fittings, (2) profile/column retrieval of aerosol, and (3) profile/column retrieval of trace gases. Irie et al. (2008a, 2008b, 2011, 2015) described the retrieval procedures, and the error estimates. Herein we provide a short overview.

First, the differential slant column density (ΔSCD) of trace gases is retrieved using the DOAS technique (Platt 1994), which uses the nonlinear least-squares spectral fitting method, according to the following equation :

$$lnI(\lambda) = ln(I_o(\lambda) - c(\lambda)) - \sum_{i}^{n} \sigma_i(\lambda)\Delta SCD_i - p(\lambda) \qquad (1)$$

Therein, $I_o(\lambda)$ represents the reference spectrum measured at time $t$. $I_o(\lambda)$ is derived by interpolating two reference spectra (i.e., EL=70°) within 15 min before and after the complete sequential scan of the off-axis ELs at time $t$. $\Delta SCD$ represents the difference between the slant column density along the off-axis and reference spectrum. Second- and third-order polynomials are fitted to account, respectively, the wavelength-dependent offset $c(\lambda)$ and the effect of molecular and particle scattering $p(\lambda)$. In addition, $c(\lambda)$ accounts for the influence of stray light. The HCHO $\Delta SCD$ and $NO_2$ $\Delta SCD$ are retrieved respectively, from the fitting windows of 340–370 and 460–490 nm. Significant $O_4$ absorptions in the 338–370 and 460–490 nm fitting windows are used to retrieve the $O_4$ $\Delta SCDs$. The absorption cross-section data sources and the fitted absorbers in the HCHO and $NO_2$ fitting windows are given in Table 1. Figure 2 presents an example of the fitting results. $O_4$ fittings in both retrieval windows are shown in Fig S1 (supplementary information).

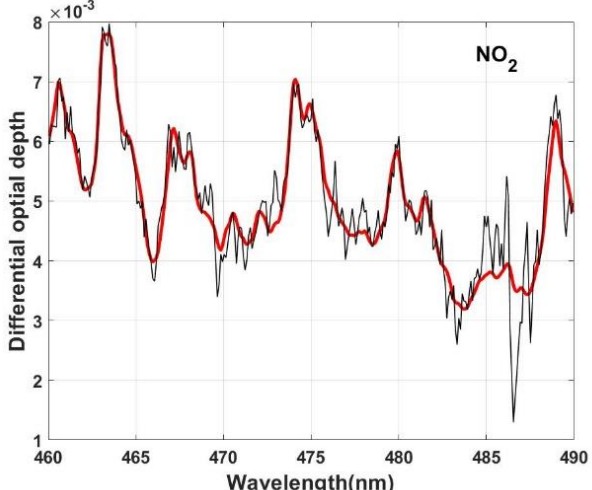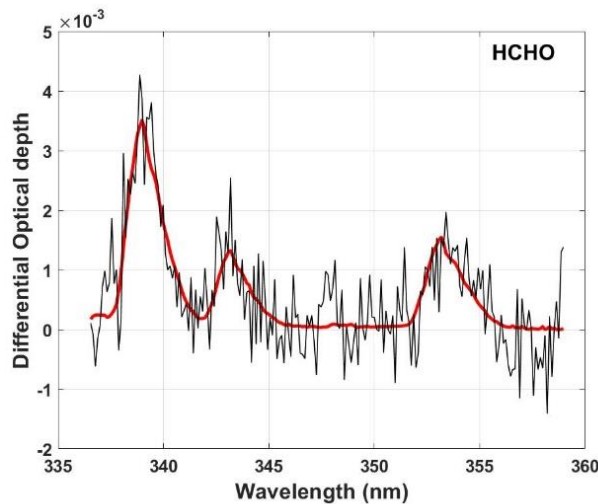

**Figure 2:** Examples of spectral fitting of NO₂ and HCHO, where red and black lines respectively show the scaled
cross-section and the summation of scaled cross-sections and fitting residuals. The example shows the
measurements of 10 April 2017, in Phimai at 10:00 LT at an EL of 2°.
In the second step, the aerosol optical depth (AOD) $\tau$ and the vertical profiles of the aerosol extinction
coefficient (AEC) $k$ are retrieved using the approach developed by Irie et al, (2008a) which is based on
the optimal estimation method (Rogers, 2000). In this approach, the measurement vector $y$ (representing
the quantities to be fitted) and state vector (representing the retrieved quantities) is defined as

$$y = (O_4 \Delta SCD(\Omega_1) \ldots \ldots \ldots \Delta SCD(\Omega_n))^T \qquad (2) \text{ and}$$
$$x = (\tau F_1 F_2 F_3)^T \qquad (3),$$

**Table 1.** Cross-section data references and absorbers fitted in the HCHO and $NO_2$ windows

| Cross-section | Absorbers fitted | Data Source |
|---|---|---|
| $O_3$ | | Bougmil et al. (2003), 223K |
| $NO_2$ | $O_3$, $NO_2$, $H_2O$, $O_4$, Ring | Vandaele et al. (1996), 295K |
| BrO | | Fleischmann et al. (2004), 223K |
| Ring | | Chance and Spurr (1997) |
| $H_2O$ | | Vandaele et al. (2005), 280K |
| $O_4$ | | Hermans et al. (2003), 296K |
| HCHO | $O_3$, $NO_2$, HCHO, BrO, $O_4$, Ring | Meller and Moortgart (2000), 293k |

where $n$ stands for the number of measurements within one complete scan of an EL sequence. Also, $\Omega$
denotes the viewing geometry and includes three components: solar zenith angle (SZA), EL, and relative
azimuth angle (RAA). The $F$ values determine the profile shape, with values between 0 and 1. The partial
AOD for 0–1, 1–2, 2–3, and above 3 km layers were defined respectively  as $AOD \cdot F_1$, $AOD \cdot (1-F_1) F_2$,
and $AOD \cdot (1-F_1)(1-F_2) F_3$, and $AOD \cdot (1-F_1)(1-F_2)(1-F_3)$. The AEC profile from 3 to 100 km is derived
assuming a fixed value at 100 km and exponential AEC profile shape with a scaling height of ~1.6 km.
The k value at 100 km was estimated from Stratospheric Aerosol and Gas Experiment III (SAGE III)
aerosol data ($\lambda$=448 and 521 nm) taken at altitudes of 15–40 km. The negligible influence of such
assumptions on the retrievals in the lower troposphere has been demonstrated in sensitivity studies
reported by Irie et al (2012). Similarly, the AEC profiles at 2–3, 1–2, and 0–1 km were derived. Such
parameterization provides the advantage that the AEC profile can be retrieved using only the apriori
knowledge of the $F$ values (profile shape) and little or no information related to the absolute AEC values
in the troposphere. Irie et al. (2008a) demonstrated that the relative variability of the profile shape, in
terms of 1-km averages, is smaller than that of the absolute AEC values. AEC profile shapes
corresponding to different $F$ values is shown in Fig.S2 (supplementary information). However, the
vertical resolution and the measurement sensitivity cannot be derived directly with such a
parameterization (Irie et al., 2008a; 2009). The retrievals and simulations conducted by other groups for
similar geometries (i.e., Frieß et al., 2006) are used to overcome such limitations. The apriori values used
for this study were similar to those reported by Irie et al. (2011): AOD = 0.21 ± 3.0, $F_1$ = 0.60 ± 0.05, $F_2$
= 0.80 ± 0.03, and $F_3$ = 0.80 ± 0.03.
Then, a lookup table (LUT) of the box air mass factor ($A_{box}$) vertical profile at 357 and 476 nm is
constructed using the radiative transfer model JACOSPAR (Irie et al., 2015), which is based on the Monte
Carlo Atmospheric Radiative Transfer Simulator (MCARaTS) (Iwabuchi, 2006). The values of the single-
scattering albedo (s), asymmetry parameter (g), and surface albedo were, respectively, 0.95, 0.65 (under
the Henyey-Greenstein approximation), and 0.10. The U.S. standard atmosphere temperature and pressure
profiles were used for radiative transfer calculations. Uncertainty of less than 8% related to the usage of
fixed values of s, g, and a were estimated from sensitivity studies (i.e., Irie et al 2012). Results obtained
from JACOSPAR are validated in the study reported by Wagner et al. (2007). The optimal aerosol load
and the $A_{box}$ profiles are derived using the $A_{box}$ LUT and the $O_4$ $\Delta$SCD at all ELs.
In the third step, the $A_{box}$ profiles, HCHO and $NO_2$ $\Delta$SCDs, and the nonlinear iterative inversion
method are used to retrieve the HCHO and $NO_2$ vertical column densities (VCDs) and profiles. Here the
$NO_2$ retrieval is explained.

For trace gas retrieval, the measurement vector and state vector are defined as

$$y = (NO2\Delta SCD(\Omega_1) \ldots\ldots\ldots NO2\Delta SCD(\Omega_n))^T \qquad (4) \text{ and}$$

$$x = (VCDf_1f_2f_3)^T \qquad (5)$$

VCD represents the vertical column density below 5 km. The $f$ values are the profile shape factors. Above the 5 km layer, fixed profiles are assumed. Similarly, to aerosol retrieval, the partial VCD values for the 0–1, 1–2, 2–3, and 3–5 km is defined respectively as VCD $\cdot f_1$, VCD $\cdot (1\text{-}f_1) f_2$, VCD $\cdot (1\text{-}f_1) (1\text{-}f_2) f_3$, and VCD $\cdot (1\text{-}f_1) (1\text{-}f_2) (1\text{-}f_3)$. Finally, the volume mixing ratio (VMR) is calculated using the partial VCD, and U.S. standard atmosphere temperature and pressure data scaled to the respective surface measurements.

The calculated vertical profile is converted to $NO_2$ $\Delta$SCDs using the $A_{box}$ LUT constructed for aerosol retrieval. However, the trace gas wavelengths differed from the representative wavelengths of $A_{box}$ LUT (357 and 476 nm). Therefore, the AOD at the trace gas wavelength is estimated, converting the retrieved AOD to the closer aerosol wavelength of 357 or 476 nm, assuming the Angstrom exponent value of 1.00. The choice of the Angstrom exponent value can induce uncertainty in the retrieved VCDs. However, such uncertainty was found to be non-significant compared to that of $A_{box}$ profiles. Uncertainty in the $A_{box}$ profiles are assumed to as high as 30 to 50%. Such values are derived empirically from comparison with sky radiometer and LIDAR observations (i.e., Irie et al., 2008b). Then, the $A_{box}$ profiles from the LUT corresponding to the recalculated AOD values are selected. The dependence of the $A_{box}$ profiles on the concentration profiles is expected to be low because both HCHO and $NO_2$ are optically thin absorbers (Wagner et al., 2007; Irie et al., 2011). For every 15 min (time necessary for one complete scan of ELs), 20% (the mean ratio of the retrieved VCD to maximum $\Delta$SCD) of the maximum trace gas $\Delta$SCDs is used as a priori information for the VCD retrievals. The a priori error is set to 100% of the maximum trace gas $\Delta$SCD. Figure 3 presents the mean averaging kernel (AK) of the HCHO and $NO_2$ retrievals during the dry season at Phimai. The area (Rodgers, 2000) provides an estimate of the measurement contribution to the retrieval. The total area is the sum of all the elements in the AK and weighted by the a priori error (Irie et al. 2008a). The areas for VCD and $f1$ of $NO_2$ retrieval are 1 and 0.6, respectively. The $f2$ and $f3$ values are much smaller. Consequently, at first, the a priori profiles were scaled, and later $f$ values determined

the profile shape. The VCD area is close to unity, and therefore, the retrieved VCD is independent of the
a priori values. Irie et al (2008) conducted sensitivity studies of choice of the $f$ values and reported
negligible effect on the retrievals.

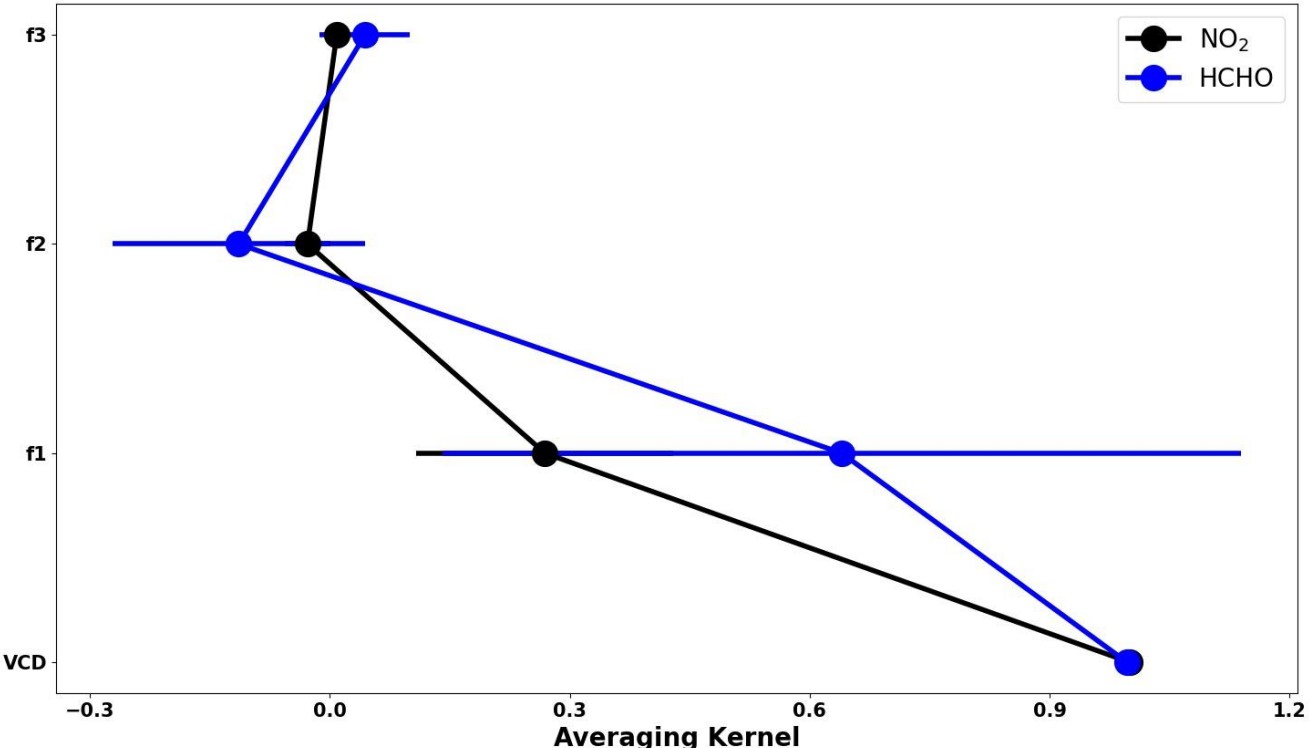


**Figure 3:** Mean averaging kernel of the $NO_2$ and HCHO retrievals from observations at Phimai during 2017. The
error bars represent the 1-sigma standard deviation of the mean values.

The total error of the retrieval consists of random and systematic errors. The measurement error
covariance matrix constructed from the residuals of the respective trace gas $\Delta$SCDs is used to estimate
the random error. The systematic error is calculated while assuming uncertainties as high as 30 and 50%
in the retrieved AOD (or the corresponding $A_{box}$ values). Table 2 shows the total estimated error. Aside
from the random and systematic error, more sources of error might exist. For instance, the bias in the ELs
can induce uncertainties in the retrieved products. However, Hoque et al. (2018) demonstrated that such
biases had a non-significant effect on the final retrieved products, mostly less than 5%.
The cloud screening procedure is similar to that described by Irie et al. (2011) and by Hoque et al.
(2018a, 2018b). During the retrieval steps, retrieved AOD values greater than 3 are excluded, because
optically thick clouds are primarily responsible for such large optical depth. Filtering based on the
residuals of $O_4$ and the trace gas $\Delta$SCDs is also used to screen clouds. Larger residuals likely occur due
to two reasons: (1) when the constructed profile is too simple to represent the true profile, particularly
with a steep vertical gradient of extinction due to clouds, and (2) rapid changes in optical depth within 30
min (time for one complete scan) (Irie et al, 2011).The screening criteria are: respective residuals of $O_4$,
HCHO, and $NO_2$ $\Delta$SCDs < 10%, < 50%, and <20%, and the degrees of freedom of retrievals greater than
1.02. The threshold values were determined statistically corresponding to the mode plus one sigma (1$\sigma$)
in the logarithmic histogram of relative residuals.

**Table 2.** Estimated Errors (%) for the $NO_2$ and HCHO concentration in 0-1 km layer, retrieved using the
JM2 algorithm

| Retrieved Product | Random error | Systematic error | Error related to instrumentation | Total error |
|---|---|---|---|---|
| NO₂ | 10 | 12 | 5 | 16 |
| HCHO | 16 | 25 | 5 | 30 |



## 2.3 CHASER simulations
CHASER 4.0 (Version 4) (Sudo et al., 2002; Sudo and Akimoto, 2007; Sekiya and Sudo, 2014), coupled
online with the MIROC-AGCM atmospheric general circulation model (AGCM) (K-1 model developers,
2004) and the SPRINTARS aerosol transport model (Takemura et al., 2005, 2009), is a global chemistry
transport model used to study the atmospheric environment and radiative forcing. In addition, several
updates, including the introduction of aerosol species (sulfate, nitrate, etc.) and related chemistry,
radiation, and cloud processes, have been implemented in the latest version of CHASER.
CHASER can calculate the concentrations of 92 species through 263 chemical reactions (gaseous,
aqueous, and heterogeneous chemical reactions) considering the chemical cycle of $O_3$–$HO_x$ – $NO_x$ –$CH_4$–
CO along with oxidation of non-methane volatile organic compounds (NMVOCs)(Miyazaki et al., 2017).
The chemical mechanism is largely based on the master chemical mechanism (MCM,
http://mcm.york.ac.uk)(Jenkin et al., 2015). CHASER simulates the stratospheric $O_3$ chemistry
considering the Chapman mechanisms, catalytic reactions related to halogen oxides ($HO_x$, $NO_x$, $ClO_x$,
and $BrO_x$), and polar stratospheric clouds (PSCs). Resistance-based parameterization (Wesely, 1989),
cumulus convection, and large-scale condensation parameterizations are used to calculate dry and wet
depositions. The piecewise parabolic method (Colella and Woodward, 1984)
**Table 3:** Settings of the CHASER simulations used in this study

| Simulation | Anthropogenic emissions | Pyrogenic emissions | Biogenic emissions | Soil $NO_x$ emission | Other physical and chemical processes |
|---|---|---|---|---|---|
| **Standard** | ON | ON | ON | ON | ON |
| **L1_HCHO** | ON | Pyrogenic VOCs switched | ON | ON | ON |
| **L1_opt** | ON | OFF | Reduced by 50% | ON | ON |
| **L1_NO₂** | ON | ON | ON | OFF | ON |

| L2 | Anthropogenic VOC emissions switched OFF | ON | ON | ON | ON |

---

and the flux-form semi-Lagrangian schemes (Lin and Rood, 1996) calculate advective tracer transport.
CHASER simulates tracer transport on a sub-grid scale in the framework of the prognostic Arakawa–
Schubert cumulus convection scheme (Emori et al., 2001) and the vertical diffusion scheme (Mellor and
Yamada, 1974). In this study, CHASER simulations were conducted at a horizontal resolution of 2.8° ×
2.8°, with 36 vertical layers from the surface to ~50 km altitude and a typical time step of 20 min. The
meteorological fields simulated by MIROC-AGCM were nudged toward the six-hourly NCEP FNL
reanalysis data at every model time step.
The anthropogenic, biomass burning, lightning, and soil emissions of $NO_x$ were incorporated into
CHASER simulations. Anthropogenic emissions were based on HTAP_v2.2 for 2008. Biomass burning
and soil emissions from the ECMWF/MAC (Global Fire Assimilation System (GFAS)) reanalysis were
used. The biogenic emissions for VOCs are based on the process-based biogeochemical model the
Vegetation Integrative SImulator for Trace gases (VISIT) (Ito and Inatomi, 2012) simulations. The $NO_x$
production from lightning is calculated based on the parameterization of Price and Rind (1992) linked to
the convection scheme of the AGCM (Sudo et al., 2002). Isoprene, terpene, acetone, and ONMV
emissions estimates in the VISIT inventory during July were $2.14 \times 10^{-11}$, $4.43 \times 10^{-12}$, $1.60 \times 10^{-12}$, and
$9.93 \times 10^{-13}$ $kgCm^{-2}s^{-1}$. Global $NO_x$ emissions of 43.80 $TgNyr^{-1}$ are used in the simulations, considering
industries (23.10 $TgNyr^{-1}$), biomass burning (9.65 $TgNyr^{-1}$), soil (5.50 $TgNyr^{-1}$), lightning (5 $TgNyr^{-1}$),
and aircrafts (0.55 $TgNyr^{-1}$) as significant sources. Global isoprene emissions from vegetation were set to
400 $TgCyr^{-1}$.

NO$_x$ emissions in India were estimated as 14 Tg/yr in 2016, almost two-fold increase since 2005 (~8 Tg/yr), with the energy and transportation sector being the largest contributor (Sadavarte et al 2014). Indian anthropogenic non-methane VOCs (NMVOCs) emissions in 2010 were estimated ~ 10 Tg/yr , with respective contributions of  60, 16, and 12% from residential, solvents, and the transport sector( Sharma et al 2015). In Japan, vehicular exhausts (14 - 25%), gasoline vapor (9 - 16%), liquefied natural gas (7 - 10%), and liquefied petroleum gas (49 - 71%) contribute to the total VOC concentrations (Morino et al., 2011), with annual NMVOC emission of ~2 Tg (Kannari et al., 2007). Annual NO$_x$ emissions in Japan and Thailand in 2000 was estimated as ~2000 and 591 kt/yr, with the largest contribution from transport-oil use, followed by the energy and industrial sector (Ohara et al., 2007). Annual anthropogenic VOC emissions in Thailand are approximately   0.9 Tg, with 43, 38, and 20% contributed, respectively, from industrial, residential and transportation sectors (Woo et al; 2020).

Multiple CHASER simulations with different settings used for sensitivity studies are presented in Table 3.

## 2.4 Satellite observations

Tropospheric NO$_2$ and HCHO retrievals from the Ozone monitoring Instrument (OMI) were also used to evaluate the model simulations. The ultra-violet nadir-viewing spectrometer OMI, on board the Aura satellite measures backscattering solar radiation covering the spectral range of 270 – 500 nm (Levelt et al., 2006). In an ascending sun-synchronous polar-orbit, OMI crosses the equator at 13:40 LT (local time (Zara et al., 2018). OMI measures at a spatial resolution of $13 \times 24$ km$^2$ and provides daily global coverage of various trace gases including NO$_2$ and HCHO. The NO$_2$ and HCHO datasets were obtained respectively from the TEMIS (www.temis.nl, last accessed on 2022/04/23) and aeronomie (https://h2co.aeronomie.be/, last accessed 2022/05/03) websites. NO$_2$ tropospheric columns retrieved using the DOMINO version 2.0 (Boersma et al., 2011) algorithm were used for the analysis. Data meeting the following criteria were selected: cloud fraction < 0.5, SZA < 70°, surface albedo < 0.3, quality flags =0, and cross-track quality flags= 0. For HCHO, we used the BIRA-IASB v14 (De Smedt et al 2015)

retrieved products. The data filtering criteria was the following: cloud fraction < 0.4, SZA<70˚, AMF >
0.2, quality flag=0, and cross-track quality flag =0.


## 3 Results and discussion

### 3.1 Results from MAX-DOAS observations

### 3.1.1 HCHO seasonal variation

The monthly mean HCHO mixing ratios in the 0–1 and 0–2 km layers from January 2017 – December
2018 and the corresponding one sigma (1σ) standard deviations indicating the variation ranges for the
three sites are presented in Fig. 4. The HCHO levels at the Phimai site show a consistent seasonal cycle,
characterized by high VMRs during the dry season. Such enhancement is related to the influence of
biomass burning during the dry season, which has been well documented in the work of Hoque et al.
(2018). The HCHO mixing ratio at Phimai reach a peak in March or April, with a maximum of 4 – 6
ppbv. The variation in the peak concentration and timing depends mainly on the intensity of biomass
burning activities. During the wet season, the HCHO concentrations are mostly within 2–3 ppbv,
indicating a two-fold increase in HCHO abundances during the dry season. The daily mean HCHO
amounts (0 –1 km) are 0.78 - 9.84 ppbv, representing seasonal modulation of 134%.

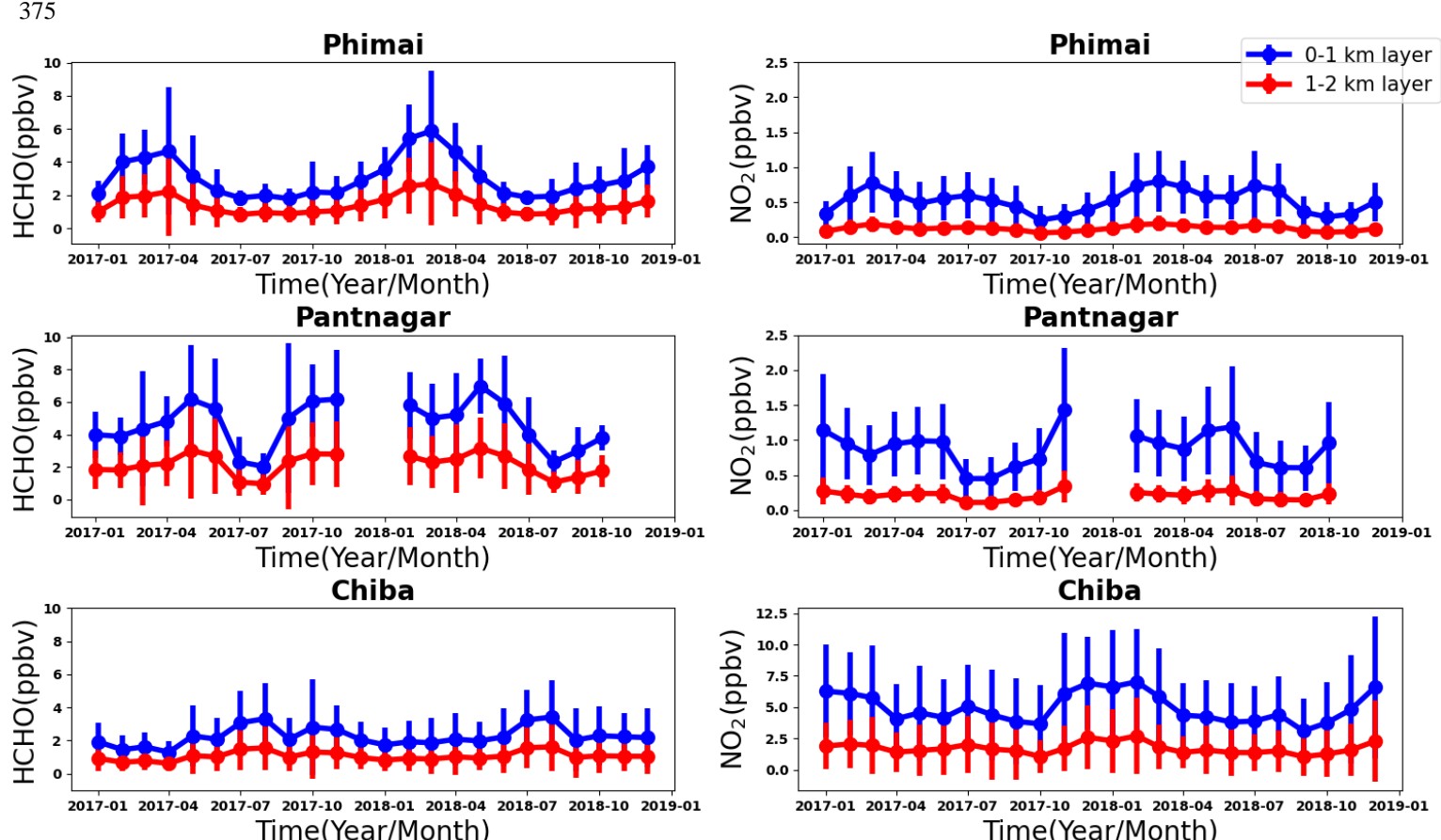

**Figure 4:** Seasonal variations in the HCHO (left panel) and NO$_2$ (right panel) mixing ratios in the 0 - 1 (blue) and 1 - 2 (red) km layers at Phimai, Pantnagar, and Chiba. The error bars represent the one sigma standard deviation of the mean values. The gaps in the plots for the Pantnagar site indicate the unavailability of observations during the investigated period.

Seasonal variation of HCHO in the 0–1 km layer at the Pantnagar site has been elucidated by Hoque et al. (2018b). Here, the results are replotted to verify the consistency of the seasonal variations. Observations made during autumn 2018 were not available because of problem with the spectrometer. Consistent seasonal variation of HCHO abundances is observed at the Pantnagar site, with enhanced concentrations during the spring. The Pantnagar site is affected by biomass burning during spring and autumn (Hoque et al., 2018b), explaining the high mixing ratios found during spring. In both years, the

maximum HCHO mixing ratios are ~6 ppbv. The springtime peak occurred in May. The HCHO concentrations during the monsoon are ~35% lower than in the spring, indicating a strong effect of the monsoon on the HCHO concentrations found for Pantnagar. The seasonal modulation of HCHO at Pantnagar estimated from the daily mean concentrations is 107%. The peak HCHO mixing ratio at Pantnagar is almost twice that of in Pune city (~ 3 ppbv) (Biswas and Mahajan, 2021), a site in the IGP region. The HCHO seasonality at the two sites are found to be dissimilar, because of differences in the VOC sources, however, lower mixing ratios during the monsoon is consistent. From another site in the IGP region (i.e., Mohali), Kumar et al., (2020) reported lowest HCHO VCDs during March 2014 and 2015, attributing them to lower biogenic and anthropogenic VOC emissions. At Pantnagar, the lowest HCHO mixing ratios are observed during the monsoon. The rainfall events in the IGP region shows strong annual variability (Fukushima et al. 2019). Discrepancies between the sites might be related to the rainfall pattern.

Under the influence of biomass burning, the maximum monthly HCHO mixing ratios at Phimai and Pantnagar are similar (~6 ppb). The maximum instantaneous HCHO VMR during biomass burning influence in Phimai and Pantnagar are, respectively, 26 and 30 ppbv. Zarzana et al. (2017) reported HCHO abundances of ~60 ppbv in fresh biomass plumes in the US. The lower values obtained from our measurements might be attributable to (1) more aged plumes intercepted by the MAX-DOAS instruments and (2) differences in the types of biomass fuel used. Comparison to reports of literature indicates that the retrieval of HCHO under biomass burning is reasonable.

The summertime maximum and wintertime minimum characterize the seasonal variations of HCHO at the Chiba site, with a peak at ~3 ppbv. The HCHO concentrations are ~2 ppbv during other seasons, which are similar to the HCHO concentrations in Phimai during the wet season. The seasonal variation amplitudes of HCHO at Chiba is ~94%. For a site with similar seasonal variation (i.e., summertime maximum and wintertime minimum), Franco et al. (2015) reported HCHO seasonal modulation of 88%.

The HCHO VMRs in the 1–2 km layers at all three sites are lower, almost 50% the value of the concentrations in the 0–1 km layer. The HCHO seasonal variation amplitudes at Phimai, Pantnagar, and Chiba sites are, respectively, 131%, 102%, and 90% when calculated based on the HCHO concentration in the 1–2 km layers. The modulation was even lower when retrieved values for the 2–3 km layer is used.

### 3.1.2 NO₂ seasonal variation at the three sites

Figure 4 also shows the seasonal variation of $NO_2$ in the 0–1 and 1–2 km layers at the three sites. The error bars represent the 1σ standard deviation of the mean values. The $NO_2$ seasonal variations at Phimai and Pantnagar sites are similar to those of HCHO. Pronounced peaks attributable to biomass burning influence is observed during the dry season at Phimai (~0.8 ppbv) and during spring (1.2 ppbv), and post-monsoon (1.4 ppbv) at Pantnagar. The lowest $NO_2$ mixing ratios at Phimai and Pantnagar are, respectively, ~0.2 and 0.5 ppbv. The $NO_2$ VMRs at Chiba is higher (~7 ppbv) during winter. The longer lifetime of $NO_x$ and lower $NO/NO_2$ ratio because of lower photochemical activity in winter lead to high $NO_2$ mixing ratios at Chiba (Irie et al., 2021).

At Phimai, the $NO_2$ mixing ratios in both seasons are similar. However, when Hoque et al. (2018a) reported the seasonal variations in $NO_2$ at Phimai during 2015 – 2018, the dry season mixing ratios were higher. Table 4 shows the number of fire events during the dry seasons during 2015 - 2018. The fire data are extracted from the MODIS Active Fire Detections database (https://firms.modaps.eosdis.nasa.gov, last accessed on 2021/12/15). Data fulfilling the following criteria were chosen – (a) data points located within 100 km of the Phimai site, (b) confidence of the data greater than 70%, and (c) observations during the daytime. The lower fire counts during 2017 - 2018 compared to those of 2015 - 2016 period coincide with the lower $NO_2$ levels in the former. Fire counts varied between 2017 and 2018 but did not affect the $NO_2$ levels. However, HCHO levels changed with the number of fire occurrences between 2015 – 2018 (i.e., Figure 4 and Hoque et al.,2018a).

At such low $NO_2$ levels at Phimai, soil $NO_x$ emissions are likely to make a greater contribution to $NO_2$. Although $NO_2$ is not emitted directly from soils, biological processes emit NO, which is rapidly converted to $NO_2$ (Hall et al., 1996). In addition, many studies have established a relation between soil moisture and NO emissions (Carden et al., 1993; Zheng et al., 2000; Schindlbacher et al., 2004; Huber et al., 2020). The potential contribution of soil $NO_x$ emissions, as inferred from CHASER simulations, is discussed in section 3.4.2.

**Table 4:** Number of fire events occurring during the dry season (January - April) at Phimai from 2015 - 2018. Selection criteria of the data are the following: (1) situated within 100 km of the site, (2) confidence level > 70%, and (c) daytime measurements.

| Dry season years | Number of fire events |
|---|---|
| **2015** | 84 |
| **2016** | 98 |
| **2017** | 62 |
| **2018** | 77 |



**3.1.3.1 The HCHO to NO$_2$ ratio ($R_{FN}$):**
The HCHO to NO$_2$ ($R_{FN}$) ratio is regarded as an indicator of high ozone O$_3$ sensitivity (Martin et al., 2004;
Duncan et al., 2010). The O$_3$ production regime is characterized as VOC-limited for $R_{FN} < 1$ and NO$_x$-
limited when $R_{FN} > 2$, and the values in the range 1-2 are said to be in the transition/ambiguous region
(Duncan et al., 2010; Ryan et al., 2020). Subsequent to a report of Tonnesen and Dennis (2000), several
studies used $R_{FN}$ estimated from satellite and ground-based observations to infer O$_3$ sensitivity to NO$_x$
and VOCs (Martian et al., 2004; Duncan et al., 2010; Jin and Holloway et al.,2015; Mahajan et al., 2015;
Irie et al.,2021; etc.). However, the effectiveness of $R_{FN}$ is still under discussion primarily based on two-
points- (1) the range of the transition region to categorize the VOC and NO$_x$ -limited region, and (2) the
altitude dependence of $R_{FN}$ (Jin et al., 2017). Most of the studies described above used the transition range
($1 < R_{FN} < 2$) proposed by Duncan et al. (2010). Schroeder et al. (2017) reported that a common transition
(i.e., $1 < R_{FN} < 2$) range might not be valid globally. Instead, it should be calculated based on the region.
First, the results based on the standard transition range are discussed herein, and then its applicability to
the study regions is assessed.
Figure 5 shows scatter plots of the daily mean NO$_2$ and HCHO concentrations in the 0 - 2 km layer at the
three sites, color-coded with the respective O$_3$ concentrations (0-2 km). Retrieval of the JM2 O$_3$ product
is explained by Irie et al. (2011). The O$_3$ concentrations for SZA $< 50°$ are used to minimize stratospheric
effects. This criterion on the SZA is also applied for the selection of the NO$_2$ and HCHO concentrations.
Although not checked here, the JM2 O$_3$ product showed good agreement with ozonesonde measurements
in Tsukuba (Irie et al., 2021). Most of the high O$_3$ occurrences fall in the $R_{FN} > 2$ region at Phimai and
Pantnagar and in $R_{FN} < 1$ at Chiba. The common transition range classifies the O$_3$ production regime as
$NO_x$-limited at Phimai and Pantnagar and VOC-limited at Chiba. At all sites, the $R_{FN}$ values tend to be
biased to a particular regime (i.e., $NO_X$ - or VOC-limited), with only 4 and 2% of the ratios in the range
0 - 2, at Phimai and Pantnagar, respectively. This finding suggests that the transition occurs at a higher or
lower ratio than the common definition. Recent report by Souri et al. (2020) found that the $NO_2$-HCHO
relation plays an important role in determining the transition region and derived a formulation from
accounting for the $NO_2$-HCHO chemical feedback in the ratios as

$$HCHO = m * (NO_2 - b) \tag{6}$$


where m and b respectively denote the slope and intercept. Equation (6) is based on observations, which
means that the regionally adjusted fitting coefficients will reflect the local $NO_2$ - HCHO relation. Solving
equation (6), the transition line estimated from the observations in the 0 - 2 km layer, is shown in Fig 5
(bottom panel). Rather than a range, the method calculates a single transition line, which corresponds to
the $NO_2$ - HCHO feedback. The regions above and below the transition line are characterized, respectively
as VOC- and $NO_x$ -limited or other.

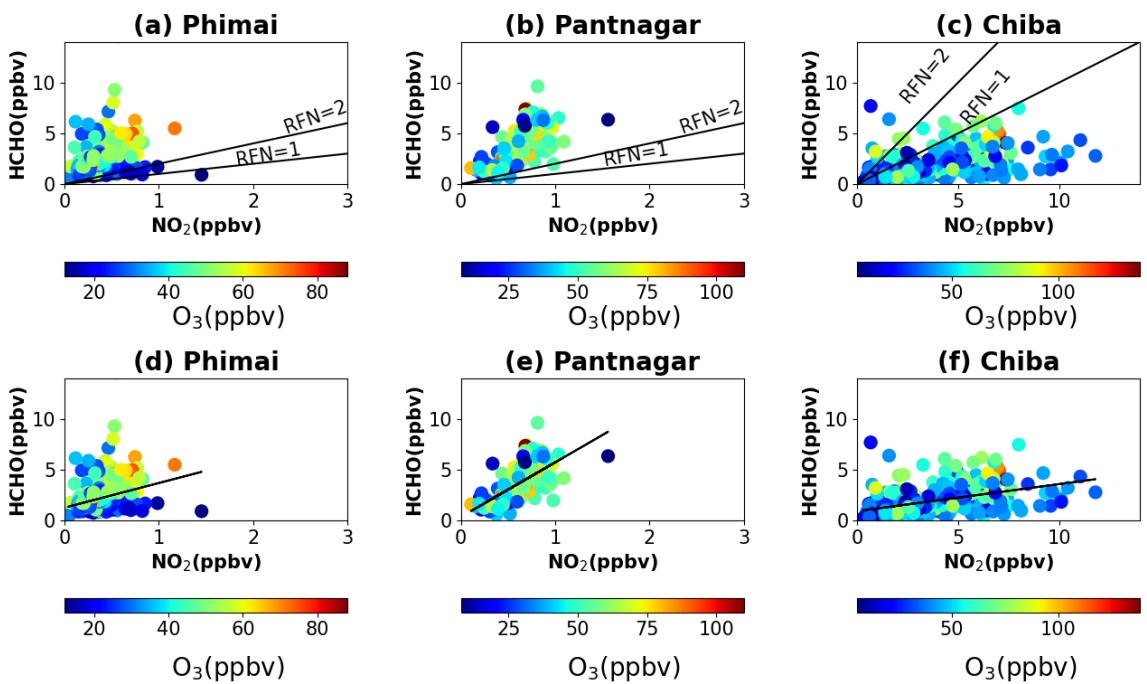


**Figure 5.** Scatter plots of HCHO and $NO_2$ concentrations in the 0-2 km layer at (a, d) Phimai, (b, e) Pantnagar, and (c, f) Chiba, coloured with the $O_3$ concentrations in the 0-2 km layer at the respective sites. The solid lines in the top panel represent $R_{FN} = 2$ and $R_{FN} = 1$ benchmarks. The black lines in the bottom panel are calculated according to equation (1).

The revised transition line at Phimai and Pantnagar is apparently more reasonable than the earlier method. At Phimai, the transition line almost clearly distinguishes between the high and low $O_3$ occurrences. It is perceptible that when the HCHO concentrations are higher than $NO_2$, the transition of the regimes is likely to occur at higher $R_{FN}$ values. The minimum and mean $R_{FN}$ value along the transition line are 3.62 and 6.78, respectively. Because Phimai is a VOC-rich environment, the regime transition occurs at higher $R_{FN}$ values than by the conventional definition. This finding echoes the results reported by Schroeder et al. (2017) for a regionally variable transition region. The definition of $R_{FN} < 1$ as a VOC -limited regime might not be valid in this case. Considering the mean $R_{FN}$ ratio along the transition line (i.e., 6.78), the VOC- and $NO_x$ -limited (and other) regimes are defined, respectively as $R_{FN} < 6.78$ and $R_{FN} > 6.78$. Based on this definition, around 34% (65%) of the ratios are higher (lower) than 6.78, classifying Phimai as a dominant VOC-limited region, which contradicts earlier results. Biomass burning affects Phimai during January - April and is a significant emission source in addition to biogenic emissions. Thus, high $O_3$ occurrences likely occur only 30% of the time during a year. Such events mostly lie above the transition line.

At Pantnagar, high $O_3$ occurrences lie below (42%) and above (57%) the transition line, indicating that $O_3$ production is sensitive to both HCHO and $NO_2$ which contradicts results reported by Biswas et al. (2019). Based on satellite and ground-based observations, the study estimated the $R_{FN}$ values at a site in the IGP as > 4 and >2 respectively, and regarded the $O_3$ regime as $NO_x$-limited. Mahajan et al. (2015) reported $R_{FN}$ values of less than 1 over the IGP region signifying as a VOC-limited region. Pantnagar is a sub-urban site situated beside a busy road. Therefore, effects of anthropogenic emissions are expected year-round, especially with pyrogenic emissions during the spring and post-monsoon period. $O_3$ sensitivity to both $NO_x$ and VOCs in the north-west IGP region has also been reported by Kumar and Sinha (2021). Therefore, the balance between the VOC and $NO_x$-limited region in the IGP is reasonable. The mean and minimum $R_{FN}$ value along the transition line are, respectively, 5.59 and 6.09. The minimum

value (i.e., 5.59) is higher than Phimai (3.26), suggesting higher VOC levels at Pantnagar, consistent with
the observations.
At Chiba, 60% of the $R_{FN}$ values lie below the transition line, suggesting a dominant VOC-limited
region, which is consistent with the results reported by Irie et al. (2021). The minimum and the mean $R_{FN}$
along the transition line are, respectively, 0.33 and 0.72.  The transition occurs at a low $R_{FN}$ value because
of higher $NO_2$ levels. The fact that, 40% of the $R_{FN}$ values are above the transition region suggests a
moderate effect of HCHO on the $O_3$ sensitivity at Chiba.
Although the new classification results are apparently reasonable, they should be interpreted with
care. Our current understanding of $R_{FN}$ contradicts the classification of rural sites as VOC-limited. Despite
the theoretical and observational evidence (i.e., Souri et al.,2020), the classification of regimes based on
a single transition line is not yet well-established. Schroder et al. (2017) used regionally varying transition
ranges. Moreover, (a) the number of observations and (b) the systematic and retrieval errors can affect
the estimations and classifications. These findings are expected to contribute to the ongoing discussion
about the effectiveness of $R_{FN}$. However, the results support the idea of a regionally varying transition
range.

**3.1.3.2 R$_{FN}$ profiles**
Figure 6 shows the seasonal mean $R_{FN}$ profiles at the three sites. Only the profiles during the high $O_3$
concentrations at the sites (i.e., March at Phimai, May at Pantnagar, and February at Chiba) are shown.
The $R_{FN}$ values are likely increase with height because of the lower vertical gradient of $NO_2$, than that of
HCHO (Fig.4). It is particularly interesting that, the $R_{FN}$ values are similar in the 1-2 km height under
biomass burning conditions, suggesting a small variation in the HCHO loss rate in the particular layer. At
both sites, the HCHO concentration at 1.5 km is about 3 ppbv. At Chiba, a considerable amount of $NO_2$
in the higher layers increases the ratio up to 2 km height. Beyond 2 km, the ratio variation at all sites is
opposite that found for the surface. The gradient issue of $R_{FN}$ has been discussed explicitly by Jin et al.
(2017). They proposed a conversion factor to account for gradient differences in the surface and column-
derived $R_{FN}$ values, estimating the conversion factor from the model simulated surface and column
abundances of NO₂ and HCHO. We adopt the method reported by Jin et al. (2017) for this study using
the CHASER simulated NO₂ and HCHO concentrations and vertical columns.
First, CHASER simulated near-surface NO₂ and HCHO concentrations were converted to number
density. The effective boundary layer height (E) (Halla et al., 2011; Jin et al., 2017) was estimated.

$$E_{NO_2} = \frac{NO_2\ total\ column}{NO_2\ near-surface\ number\ density} \qquad (7)$$


$$E_{HCHO} = \frac{HCHO\ total\ column}{HCHO\ near-surface\ number\ density} \qquad (8)$$

Therein, $E_{NO_2}$ and $E_{HCHO}$ respectively denote the effective boundary layer heights of NO₂ and HCHO.
In the second step, the column to surface conversion factor (F) was calculated according to the following
equation:
$$F = \frac{E_{HCHO}}{E_{NO_2}} \qquad (9)$$

The seasonal variation of F for the three A-SKY sites and the associated 1σ standard deviation of the
mean values are depicted in Fig. 7(c). The F values over East Asia reported by Jin et al. (2017) were ~2,
without marked seasonal variation. CHASER estimated F values over Chiba range between 1–2.5, which
is apparently reasonable, when compared with literature values. Values reported in literature for polluted
regions (NO₂ > 2.5 molecules cm$^{-2}$) considered simulation data for 1–2 PM, but the estimates for this
study used daytime (07:00 – 18:00) simulations.

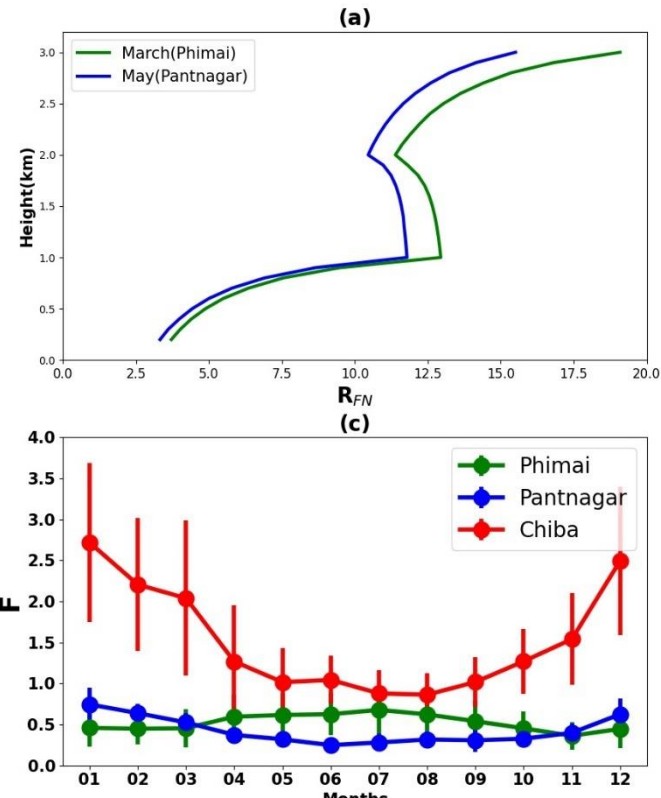

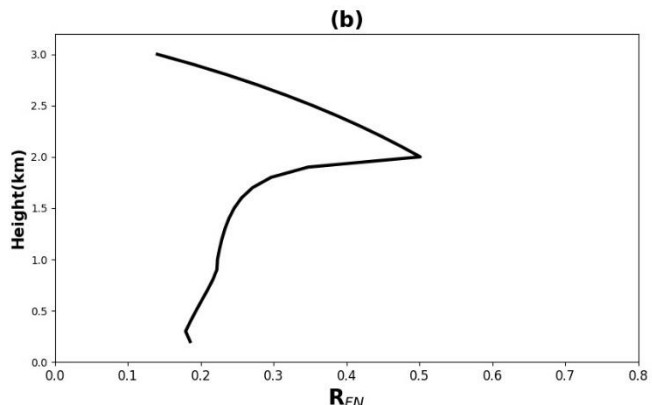


**Figure 6**: Seasonal mean $R_{FN}$ profiles during (a) March and May at Phimai and Pantnagar, respectively, and (b)
February at Chiba. (c) Seasonal variations in the column to surface conversion factor (F) for the Phimai, Pantnagar,
and Chiba sites, estimated from the CHASER simulated HCHO and $NO_2$ surface concentrations and VCD. The
simulated data from 07:00 – 18:00 in 2017 were used to estimate the F values. The error bars represent the one
sigma standard deviation of the mean values.

The F values for Pantnagar are mostly less than 1, with no distinctive seasonal variation. Mahajan et
al. (2015) reported OMI-derived $R_{FN}$ values < 1 over the IGP region. When this estimated conversion
factor is used with the values reported by Mahajan et al. (2015), the discrepancy in the satellite and
ground-based observation derived $R_{FN}$ values in the IGP region are reduced indicating that the estimated
F values for the Pantnagar site can be representative for the IGP region. The F values at the Phimai site
range were 0.5–1. Our estimated F values for the Phimai and Pantnagar sites are useful as representative
values for these respective regions, which can be improved further based on the results.

## 569 3.2 Global Evaluation of the CHASER model

This section describes the evaluation of CHASER $NO_2$ and HCHO columns for 2017 against OMI
observations. The OMI AKs were applied to the CHASER outputs to account for the altitude-dependence
of the retrievals. First, 2-houly simulated profiles ($NO_2$ and HCHO) were sampled closest to the
observation time. Secondly, AKs were applied to the sampled profiles and the mean profile was
calculated. Thirdly, both the simulations and observations were averaged on a 2.8˚ bin grid. The month
of July and December were discarded from the $NO_2$ comparison because few coincident days (only five
days) were available after filtering. It should be noted that simulations based on old $NO_x$ emission
inventory will likely affect the model-satellite comparison results. However, the current study has not
assessed such impact due to technical issues related to using an updated emission inventory. All these
issues will be addressed in a separate study.

### 580 3.2.1 Comparison between CHASER and OMI $NO_2$

Figure 7 compares the simulated and observed annual mean tropospheric $NO_2$ columns. The statistics for
the comparison are given in Table 5. The model captured the global spatial variation well with a spatial
correlation (*r*) of 0.70. The mean bias error (MBE) and the root mean square error (RMSE) are
respectively, $3 \times 10^{14}$ and $5.4 \times 10^{14}$ molecules cm$^{-2}$. On a global scale, CHASER estimations are
negatively biased by 38% compared to OMI. Actually, studies evaluating global $NO_2$ simulations with
satellite observations have reported similar negative biases (Miyazaki et al., 2012, Sekiya et al., 2018).
The differences in the spatial representativeness between the model and observations is one potential
reason for such negative biases. CHASER simulations at 1.1˚ improved the MBE and RMSE by 5 and
15%, respectively, compared to simulations at 2.8˚ (Sekiya et al. 2018). Moreover, Sekiya et al (2018)
used $NO_2$ simulations with an updated inventory and compared the results with OMI observations from
2014. Although they reported a better global spatial correlation ($r > 0.90$), the MBE ($2.5 \times 10^{14}$ molecules
cm$^{-2}$) and RMSE ($4.4 \times 10^{14}$ molecules cm$^{-2}$) values at 2.8˚ resolution are comparable to those obtained
from this study.
OMI retrievals show the highest $NO_2$ columns over eastern China (E-China) and Western Europe. Annual
mean $NO_2$ columns over the remainder of the land areas are between $7\times10^{14}$ and $4\times10^{15}$ molecules cm $^{-2}$.
Over the land areas the differences between the datasets are mostly between $-2\times10^{15}$ and $5\times10^{14}$ molecules
cm$^{-2}$. Although CHASER also underestimates $NO_2$ columns over the ocean, the differences are lower than
that of over lands. CHASER estimates are higher by $\sim5\times10^{14}$ molecules cm$^{-2}$ than OMI over Japan. Since
2012, the $NO_2$ columns have shown a declining trend over Japan, mainly because of emission controls in
China (Irie et al 2016). Probably because of simulations with an emission inventory earlier than 2012, the
simulated values tend to be higher than observations.
Figure 8 compares the seasonal variations in the monthly mean $NO_2$ columns in some selected region.
The error bars represent the 2-sigma standard deviation of the observed mean values. The numbers in
each subplot signify the regional spatial correlation between the datasets. Over eastern China (E-China),
CHASER values are negatively biased by 24%; the *r*-value is 0.68. The model captured the seasonality
well within variation range of the observations. Over E- and W-USA (eastern and western USA), the
respective *r*-values are 0.85 and 0.49 respectively. Simulated $NO_2$ columns are higher over E-USA than
over W-USA, consistent with the observations. Although, in both regions model estimates are biased by
$\sim23\%$ in the lower side compared to OMI observations, the RMSE in E-USA are $\sim40\%$ higher than in W-
USA.
Over Europe, CHASER estimates are negatively biased by 54%, with an *r*-value and RMSE of 0.80 and
$1.28 \times 10^{15}$ molecules cm$^{-2}$, respectively. The observed $NO_2$ levels over Europe are almost twice those of
the W-USA. The model was unable to capture the regional differences. Model underestimations in Europe
can be attributed to the older anthropogenic emission inventory used for the study. In fact, using the HTAP
2010 inventory the MBE ($-0.53 \times 10^{15}$ molecules cm$^{-2}$) between OMI and CHASER $NO_2$ column


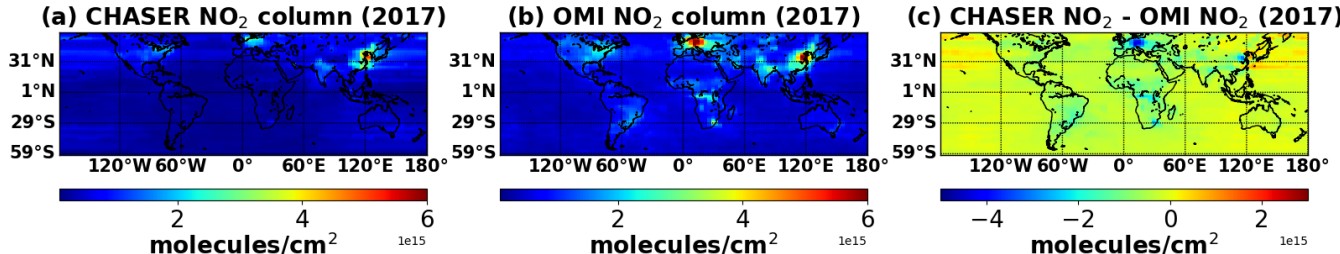

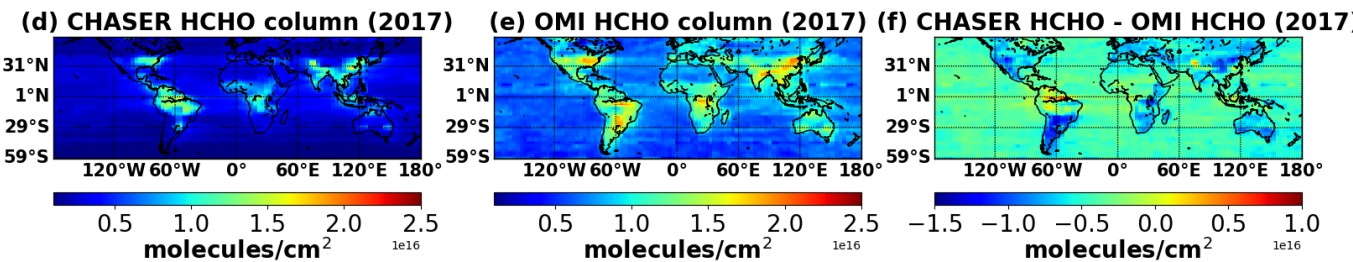

**Figure 7:** (top panel) Annual mean tropospheric $NO_2$ ($\times 10^{15}$ molecules cm$^{-2}$) columns (a) simulated by CHASER and (b) retrieved from OMI observations. Limited $NO_2$ data in July and December met the filtering criteria, thus discarded from the calculation. (c) The differences between the simulated and observed $NO_2$ columns. (bottom panel) Annual mean HCHO ($\times 10^{16}$ molecules cm$^{-2}$) columns (d) simulated by CHASER and (e) retrieved from OMI observations. (f) The differences between the simulated and observed HCHO columns. The data for 2017 are plotted only. All the datasets are mapped onto a 2.8˚ bin grid.

simulations at 2.8 over Europe (Sekiya et al. 2018) was ~ 50% lower than in the current study, although their RMSE value is similar.

Over India, MBE and RMSE for the annual mean $NO_2$ column are $-4.3 \times 10^{14}$ and $4.4 \times 10^{14}$ molecules
$cm^{-2}$, respectively, and the *r*-value is moderate (0.65). Although CHASER estimates are negatively biased
by



**Table 5:** Statistics of comparison of annual mean NO2 and HCHO columns between CHASER and OMI.MBE1
and MBE2 are the respective mean bias error. RMSE1 and RMSE2 are the respective root mean square errors. r1
and r2 signifies the respective spatial correlation coefficient. The units of MBE1 and RMSE1 are $\times 10^{15}$ molecules
$cm^{-2}$. MBE2 and RMSE2 values are in the unit of $\times 10^{16}$ molecules $cm^{-2}$.

| Region | r1 (CHASER vs OMI $NO_2$) | MBE1 (CHASER - OMI $NO_2$) | RMSE1 (CHASER – OMI $NO_2$) | r2 (CHASER vs OMI HCHO) | MBE2 (CHASER – OMI HCHO) | RMSE2 (CHASER – OMI HCHO) |
|---|---|---|---|---|---|---|
| **Global** | 0.73 | -0.30 | 0.54 | 0.74 | - 0.45 | 0.49 |
| **E-China** | 0.68 | -1.84 | 2.47 | 0.57 | -0.63 | 0.64 |
| **E-USA** | 0.85 | -0.62 | 0.63 | 0.91 | -0.56 | 0.56 |
| **W-USA** | 0.49 | -0.33 | 0.37 | 0.63 | -0.71 | 0.71 |
| **Europe** | 0.80 | -1.20 | 1.28 | 0.51 | -0.67 | 0.68 |
| **India** | 0.65 | -0.43 | 0.44 | 0.73 | -0.56 | 0.57 |
| **N-Africa** | 0.58 | -0.88 | 0.90 | 0.65 | -0.29 | 0.32 |
| **S-Africa** | 0.80 | -1.25 | 1.40 | 0.22 | -0.66 | 0.70 |
| **S-America** | 0.87 | -0.80 | 0.88 | 0.47 | -0.31 | 0.40 |
| **SE Asia** | 0.57 | -0.61 | 0.64 | 0.48 | -0.41 | 0.44 |



32%, the values lie within the 2-sigma range of the observations. Sekiya et al. (2018) found no significant
effect of higher model resolution on the MBE and RMSE in the Indian region.
Over N- and S-Africa (North and South Africa), the model values are biased low by more than 75%
compared to the observations. Prominent biomass burning occurs in both regions, which explains the
enhanced $NO_2$ levels in the OMI retrievals. High negative biases in the model values indicates that
biomass burning $NO_x$ emissions for the African regions are likely underestimated. Similarly, CHASER
underestimates $NO_2$ columns by 80% in South America, where pyrogenic emissions contributions are
significant. CHASER estimates are lower than OMI in these regions, but model captured the spatial
distribution well.
Over the SE-Asian (Southeast) region, OMI columns are enhanced during the dry season (i.e., January -
April. Burning agricultural wastes is a common practice in many countries in Southeast Asia during the
dry season, explaining the enhanced columns. The MBE (- $6 \times 10^{14}$ molecules $cm^{-2}$) and RMSE (6.4 $\times$
$10^{14}$ molecules $cm^{-2}$) in the SE-Asia region are lower than the African regions (i.e., N-Africa, S-Africa,
and S-America), where biomass burning is prominent.

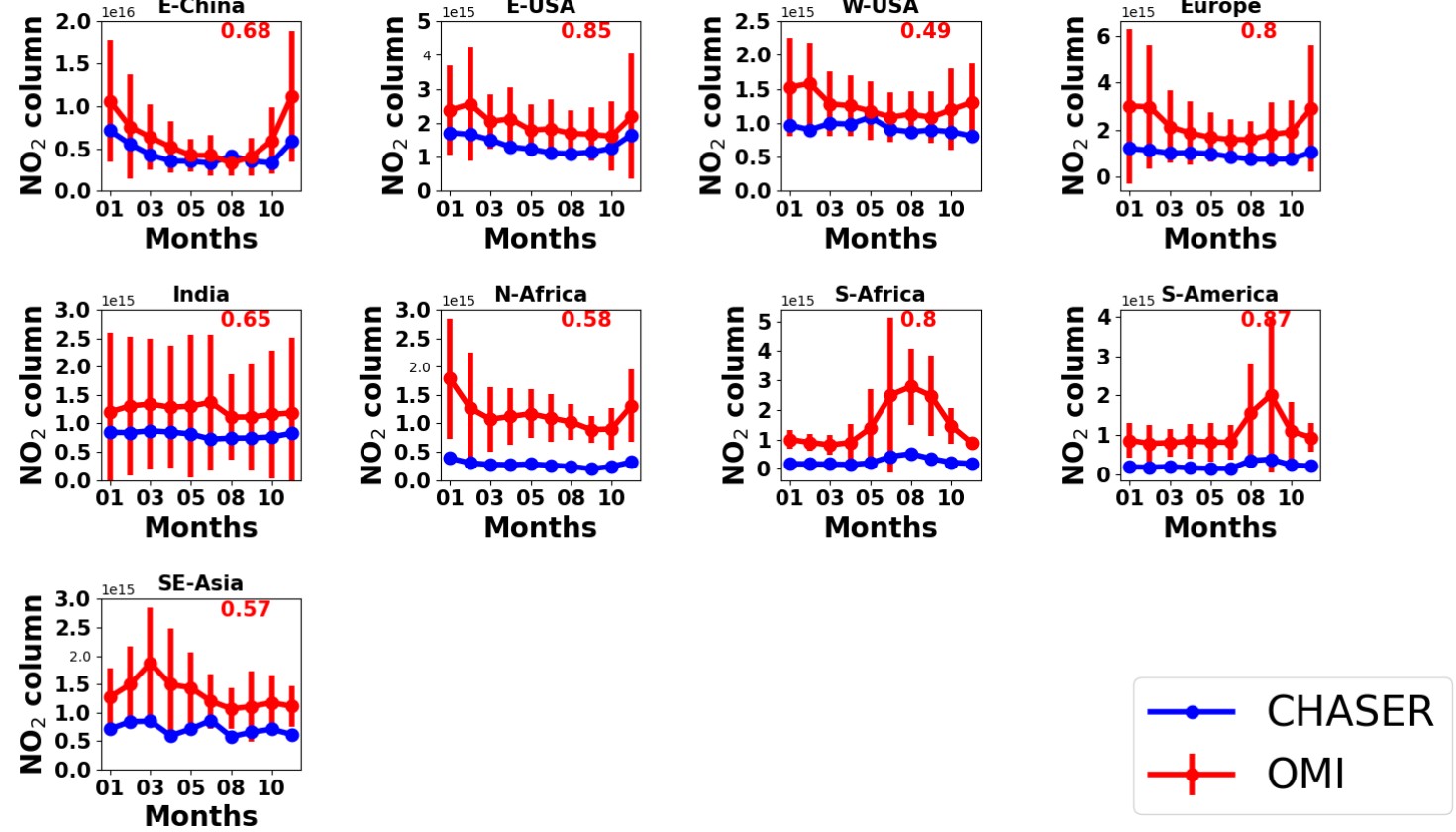


**Figure 8:** Seasonal variations in tropospheric $NO_2$ columns in E-China (110˚ -123˚ E, 30˚ – 40˚ N), E-USA (32˚ – 43˚ N, 71˚ – 95˚ W), W-USA(32˚ – 43˚ N, 100˚ – 125˚ W), Europe (35˚ – 60˚ N, 0˚ – 30˚ E), India (7.5˚ – 54˚ N, 68˚ – 97˚ E), N-Africa (5˚ – 15˚ N, 10˚ W – 30˚ E), S-Africa (5˚ -15˚ S, 10˚ -30˚ E), S-America (0˚ -20˚ S, 50˚ - 70˚ W), and SE-Asia (10˚ – 20˚ N, 9˚ – 145˚ E). CHASER simulations and OMI retrievals are plotted in blue and red colors respectively. The error bars indicate the 2-sigma variation of the observed mean values. The number in the insets signifies the regional spatial correlation between CHASER and OMI $NO_2$ columns.

## 3.2.2 Comparison between CHASER and OMI HCHO

Figure 9 presents a comparison between the simulated and observed global annual mean HCHO columns. The statistics of the comparison are given in Table 5. CHASER is able to reproduce the observed global spatial variation well with $r = 0.73$. The global MBE and the RMSE are respectively, $-4.5 \times 10^{15}$ and 4.9

$\times 10^{15}$ molecules cm$^{-2}$. MBE and RMSE for monthly mean fields show no distinctive seasonal variation
(Table S2). High HCHO columns are observed over China, Australia, Europe, India, Central Africa, South
America, and the United States. The model mostly underestimated the HCHO abundances in the higher
latitudes and Australia. Absolute differences between the model and observations in the higher latitudes
vary between $5\times10^{15}$ and $1\times10^{16}$ molecules cm$^{-2}$. Figure 9 compares the seasonal variations in the monthly
mean HCHO columns in some selected region. Therein error bars represent the 2-sigma standard
deviation of the observed mean values. The numbers in the respective subplots signify the regional spatial
correlation between the datasets.
Over E-China, CHASER HCHO estimates are negatively biased by 45% compared to OMI and the *r*-
value is greater than 0.50. The model reproduced the observed HCHO seasonality well including
enhanced peaks during the summer. The greatest differences between the datasets are observed during
the winter. Over E-USA, the spatial correlation between the datasets is greater than 0.90. Also, the
CHASER estimates are biased by 49% in the lower side. Simulations show that the peak in the HCHO
abundances occurs in July, which is consistent with the observations. The observed and simulated
magnitude of the seasonal modulation is 51 and 78%, respectively. The seasonality in the HCHO columns
in E-China and E-USA signifies a strong contribution from biogenic emissions. In both regions, the
observed peak HCHO column is ~ $1.75\times10^{16}$ molecules cm$^{-2}$. The simulated peak HCHO values are also
similar in both regions, despite the underestimation. Over W-USA and Europe, the negative biases in the
simulation are greater than 60%. However, the simulated peaks during summer are consistent with the
observations. The OMI retrievals show that the HCHO abundances in both regions are almost similar,
which has been well captured by CHASER, although the magnitude is underestimated.

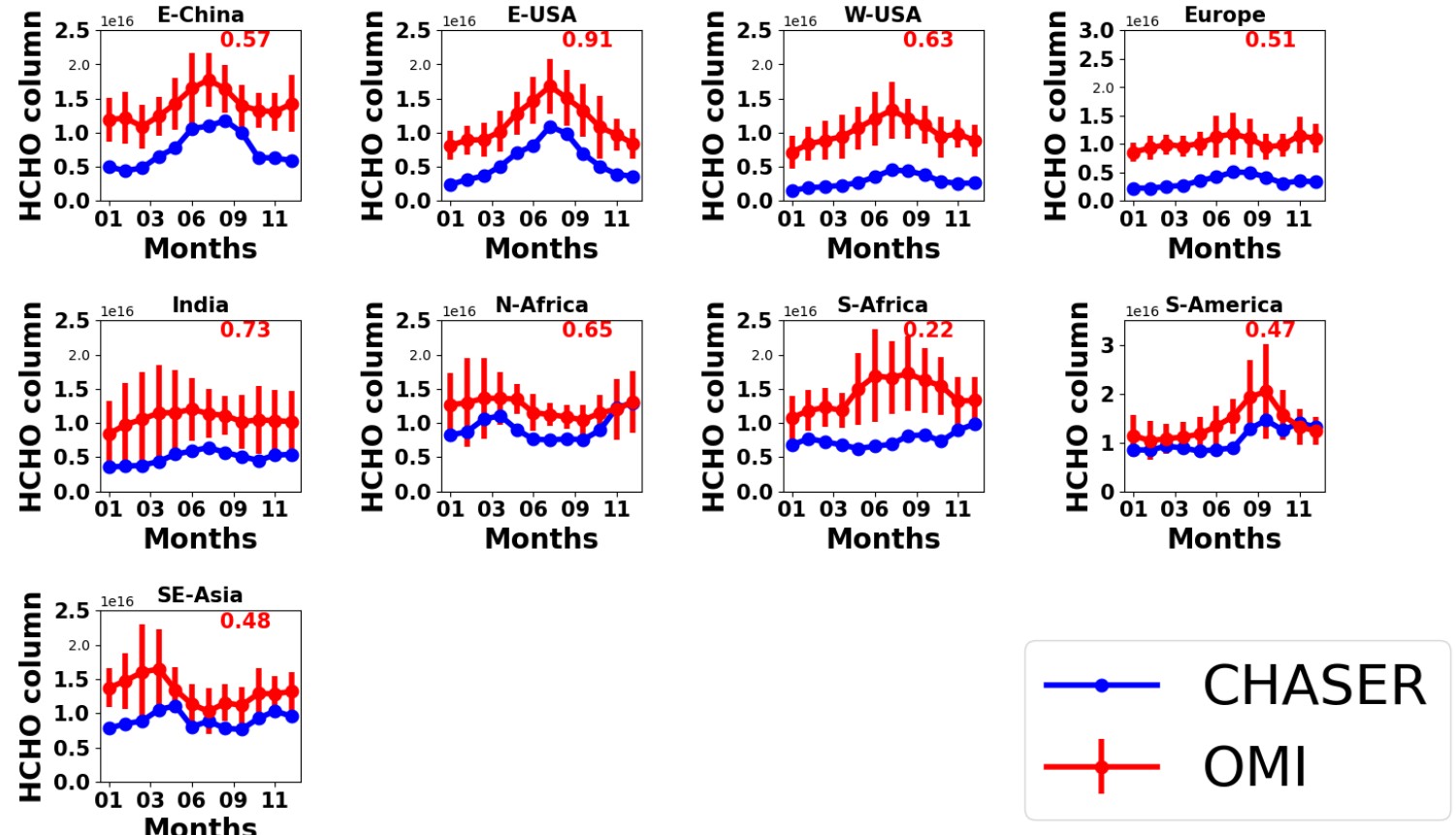


**Figure 9:** Seasonal variations in HCHO columns in E-China (110˚ -123˚ E, 30˚ – 40˚ N),  E-USA (32˚ – 43˚ N, 71˚ – 95˚ W), W-USA(32˚ – 43˚ N, 100˚ – 125˚ W), Europe (35˚ – 60˚ N, 0˚ – 30˚ E), India (7.5˚ – 54˚ N, 68˚ – 97˚ E), N-Africa (5˚ – 15˚ N, 10˚ W – 30˚ E), S-Africa (5˚ -15˚ S,  10˚ -30˚ E), S-America (0˚ -20˚ S, 50˚ -70˚ W), and SE-Asia (10˚ – 20˚ N, 9˚ – 145˚ E).  CHASER simulations and OMI retrievals are plotted in blue and red colors respectively. The error bars indicate the 2-sigma variation of the observed mean values. The number in the insets signifies the regional spatial correlation between CHASER and OMI HCHO columns.

Over India, the model estimates mostly lie outside of the observational variation ranges, although, CHASER captured the spatial distribution well ($r$ = 0.73). Magnitudes of the seasonal variation in both OMI and CHASER are around 32%.  Between the two African regions, CHASER demonstrated better capability for reproducing HCHO distribution in N-Africa ($r$ = 0.65). Negative model bias in N-Africa is

almost half (22%) that of S-Africa (46%). Observed N-African HCHO columns are mostly higher than $1.2 \times 10^{16}$ molecules cm$^{-2}$ during the biomass burning period (November - April). Although the modeled values are lower than the observed values, the year-end columns (November - December) are similar. Both datasets show low HCHO variation during May - September. Over the S-African region, the model capabilities were limited.

Over S-America, the negative bias (~22%) in the model estimates compared to the observations is similar to that of N-Africa. In addition to consistency in the year-end (November to December) columns, CHASER well reproduced the biomass burning-led enhancements. The observed and simulated magnitudes of seasonal modulation are 49 and 43%, respectively.

Over SE-Asia, CHASER reproduced the observed biomass burning-led enhanced HCHO columns during the dry season (January - April), however, the occurrence of the peak is inconsistent. As discussed in section 3.1, observed HCHO peaks related to biomass burning can vary depending on the fire numbers. The $r$-value (0.48) is moderate and model is biased by 30% in the lower side. The model negative biases in the biomass prone regions are lowest (<30%) among the discussed regions.

De Smedt et al. (2021) reported that cloud corrections can positively bias OMI HCHO columns up to 30% compared to Tropospheric Ozone Monitoring Instrument (TROPOMI) columns. Consequently, uncertainties in the observations are also likely to contribute to the observed negative biases. Comparison among CHASER, TROPOMI, and OMI HCHO columns is beyond the scope of this study. However, the effects of uncertainties in the satellite retrievals on the negative biases is discussed qualitatively and briefly. To demonstrate such effects, CHASER and TROPOMI HCHO columns for 2019 are compared in Fig S3. The simulation settings and emission inventories are similar to those explained in section 3.2.3. The comparison results are presented in Table S2. TROPOMI data has been processed following De Smedt et al. (2021). The CHASER and TROPOMI HCHO spatial distribution correlates strongly with $r$-value of 0.78. The values for MBE and RMSE are respectively, -2.3 $\times 10^{15}$ and 2.8 $\times 10^{15}$ molecules cm$^{-2}$. Compared to OMI and TROPOMI, CHASER HCHO columns are negatively biased, respectively, by 61 and 38%. The model biases are lower when compared to TROPOMI observations. Because of temporal differences in the two comparisons, the biases cannot be compared quantitatively. However, the differences in the biases signify that the observational uncertainties can strongly affect discrepancies

between the simulated and observed HCHO abundances. Moreover, using different cloud products may
introduce inconsistencies in the OMI BIRA-IASB retrievals (De Smedt et al., 2021), affecting the
comparison results. De Smedt et al. (2021) proposed to recalculate the OMI HCHO VCDS based on the
AMF information to minimize cloud-induced uncertainties. Such a detailed method will be evaluated in
our future studies.

**3.3 Evaluation of CHASER simulations at the three sites**
**3.3.1 Evaluation of CHASER HCHO at Phimai and Chiba**
The seasonally averaged observed and modeled HCHO profiles and partial columns in the 0 - 4 km
altitude range at Phimai and Chiba are presented in Fig. 10. The CHASER outputs smoothed with MAX-
DOAS averaging kernels (AK) are also depicted. The AK is applied following Franco et al. (2015). First,
the CHASER HCHO profiles are interpolated to the MAX-DOAS vertical grids. Next, the MAX-DOAS
AK information from individual retrieved profiles is seasonally averaged according to the climate
classifications of each site. Finally, the CHASER outputs on the coincident days are selected, and the
seasonally averaged AK is applied to the daily mean interpolated profile. Applying individual AKs to the
model outputs yielded similar results. The seasonally averaged AKs for both sites are shown in Fig S4.
The coincident days at Phimai and Chiba were respectively,690 and 668.
At Phimai, CHASER predicted the increase in the HCHO partial columns during the dry season and
well-reproduced the HCHO seasonality. The simulated and observed seasonality correlates strongly with
*R*-value of 0.96. The modeled monthly mean values during the dry season are found to be within the 1σ
standard deviation of the observed values, indicating that the pyrogenic emissions estimates used for the
simulations are reasonable. CHASER predicted a 41% increase in the HCHO column during January -
March, consistent with the observations (41%). CHASER overestimates the HCHO columns in both
seasons, and the mean bias error (MBE) (CHASER – MAX-DOAS) is lower ($3.7\times 10^{15}$ molecules cm$^{-2}$)
(Table 6) during the wet season. Although underestimated, the dry season smoothed column values are
within the 1σ range.

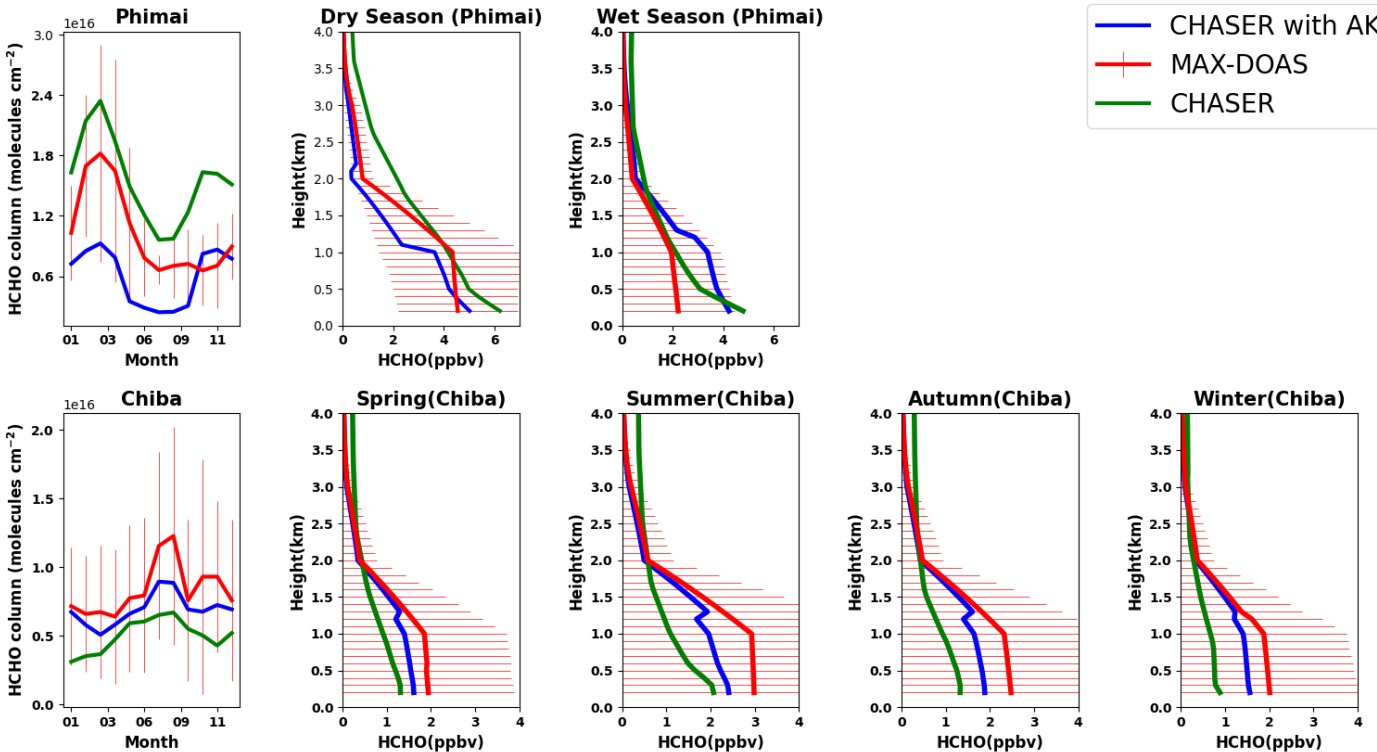

**Figure 10.** Seasonal variations in the HCHO partial columns at 0 - 4 km and vertical profiles during all seasons at Phimai and Chiba, as inferred from the MAX-DOAS observations (red) and CHASER simulation(green). The CHASER HCHO partial column and vertical profile smoothed with the MAX-DOAS AK are coloured blue. The AK information of all the screened (as explained in section 2.2) retrievals were averaged based on the seasonal classification of the respective sites. The coincident time and date between the model and observations are selected only. Error bars indicate the one sigma standard deviation of mean values of the MAX-DOAS observations.

The modeled and observed HCHO mixing ratios in the 1- 2km layers during the wet season are almost identical, whereas VMR near the surface (i.e., 0 - 1 km) differ by 30%. The absolute mean difference in the 0-4 km layer is ~0.45ppbv, with the maximum difference of 2.58 ppbv below 200 m. CHASER has demonstrated good capabilities for reproducing the HCHO profile in the 0.5 – 4 km layer during the wet season. The significance of AK information is low for the wet season. However, smoothing the model profiles reduces the overall MBE by 43%.

**Table 6:** Comparison of the seasonal mean HCHO partial columns and profiles (0-4 km) between MAX-DOAS and CHASER at Phimai and Chiba. MBE (CHASER – MAX-DOAS) is the mean bias error. The partial column and profile MBE units are respectively, $\times 10^{16}$ molecules cm$^{-2}$ and ppbv, respectively.

| Site | Season | Partial column MBE | Smoothed Partial column MBE | Profile MBE | Smoothed Profile MBE |
|------|--------|--------------------|-----------------------------|-------------|----------------------|
| Phimai | Overall | 0.28 | -0.07 | 0.35 | 0.01 |
| Phimai | Dry | 0.37 | -0.28 | 0.58 | -0.38 |
| Phimai | Wet | 0.21 | 0.07 | 0.45 | 0.33 |
| Chiba | Overall | -0.12 | -0.05 | -0.37 | -0.11 |
| Chiba | Spring | -0.07 | -0.04 | -0.22 | -0.12 |
| Chiba | Summer | -0.16 | -0.08 | -0.45 | -0.26 |
| Chiba | Autumn | -0.10 | -0.04 | -0.40 | -0.19 |
| Chiba | Winter | -0.09 | -0.01 | -0.42 | 0.11 |

During the dry season, the respective absolute mean and maximum difference in the datasets in the 0-1 km layers is ~1 and ~2ppbv. The observed and simulated seasonal differences in the 0-1 km are 50 and 34%, respectively. Simulated dry season profile values at the heights greater than ~2 km is out of the 1σ variation range. The two-potential reasons for such differences are lower measurement sensitivity in the free troposphere and the overestimated Southeast Asian biogenic emissions in the model. Despite the measurement limitations, CHASER and MAX-DOAS wet season profiles up to 3 km are consistent. Consequently, it is likely that the biogenic emissions for this region in the model are overestimated. The Southeast Asian isoprene emissions in CHASER is 128 Tgyr$^{-1}$, higher than the CMAS-GLO-BIO (Sindelarova et al., 2022) inventory (78 Tgyr-1). However, the dry season HCHO profiles in 0 - 2 km are well simulated. Smoothing underestimates the dry season profile within the 1σ variation range but improved simulations below 200 m. At heights greater than 3 km, the smoothed values mostly reproduce the a priori because of reduced measurement sensitivity (i.e., low AK value, indicating limited information was retrievable).

Moderate correlation ($R$=0.58) can be observed between the modeled and observed HCHO partial columns at Chiba. CHASER was able to reproduce the peak in the partial columns in August. The model predicts a 41% increase in the HCHO columns during January - August, whereas the observed increase is 54%. Although Chiba is an urban site, the HCHO and temperature seasonal variations show a tight correlation ($R\sim$ 0.70) (Fig S5), suggesting that changes in biogenic emissions modulate HCHO seasonality. Similarly, the modeled seasonality is consistent with temperature variation (Fig. S4). Thus, the simulated HCHO seasonality in Chiba is reasonable, despite underestimation of absolute values. Smoothing the simulations improve the correlation, and the MBE is reduced by 54% (Table 6).

The CHASER HCHO profiles in the 0 - 4 km layers are lower than the observations, with an MBE of 0.39 ppbv. The absolute differences in the modeled and retrieved HCHO profiles in the 0-2 km layer during all seasons are higher than at Phimai. Absolute mean differences of ~ 1pbbv and higher are mainly observed for 0 to 2 km. In addition, the vertical gradients of the simulated profiles are low compared to those at Phimai. The modeled profiles at Chiba resemble the HCHO profiles measured over the ocean during the INTEX-B (Intercontinental Chemical Transport Experiment: Phase B) (Boeke et al., 2011). The Chiba site is near the sea, and coarse CHASER resolution includes the ocean pixels. Moreover, urban surfaces are not homogeneous. Thus, a significant part of the profile discrepancies is likely related to the systematic differences, in addition to emission estimates. However, the model estimates lie within the standard deviation range of the measurements. Because of the low gradients in the simulated profiles, the smoothed profiles mostly imitated the a priori values even below 2 km. Overall, given the large uncertainty on the MAX-DOAS profiles (Fig. 10), the differences between the observations and smoothed profile are statistically insignificant. Effects of the horizontal resolution on the simulated HCHO levels is discussed in section 3.3.4.

### 3.3.2 Evaluation of CHASER NO$_2$ in Phimai and Chiba

Figure 11 presents the seasonal averages of the MAX-DOAS and CHASER NO$_2$ profiles and partial columns (0 - 4 km) at Phimai and Chiba. The AK is applied to the modeled outputs for the Chiba site only.

Figure S5 of the supplementary information presents a comparison of the observations, model, and
smoothed model profiles averaged within the 0 - 2 km layer at Phimai. Smoothing with different a priori
values is depicted to demonstrate the effects of the a priori values. The smoothed $NO_2$ concentrations,
calculated using the original a priori values, show a seasonal variation shift. The mean smoothed profile
resembles the observations when a priori values are reduced by 50%; however, the dry season values are
similar in both cases. Two test cases of smoothing profiles using apriori values above 500 and 800 m
shows good agreement with the observations; however, the results are sensitive to the apriori values.
Because smoothed profiles are strongly biased to the apriori choice, the smoothing results obtained for
the Phimai site are discarded.
The modeled $NO_2$ partial column at Phimai shows good agreement with observations made during the
dry season. CHASER well reproduces the enhanced $NO_2$ columns attributable to biomass burning within
the standard deviation of the observations. The peak in the $NO_2$ levels during March is consistent in both
datasets. Although the seasonality does not agree in other months, the overall MBE is $8 \times 10^{13}$ molecule
$cm^{-2}$ (Table 7). Above 500 m, the datasets shows excellent agreement. The absolute mean differences in
the 0 - 1km layer are 0.22 ppbv, and the maximum difference of ~1.9 ppbv is observed near the surface.
Amidst the biomass burning influence, the $NO_2$ concentrations at Phimai are mostly < 1 ppbv. Thus, the
results of comparisons demonstrate CHASER's good capabilities in regions characterized by low $NO_2$
concentrations. Moreover, when $NO_2$ concentrations are less than < 1 ppbv, the AK information seems
less significant if the model can capture low-concentration scenarios.

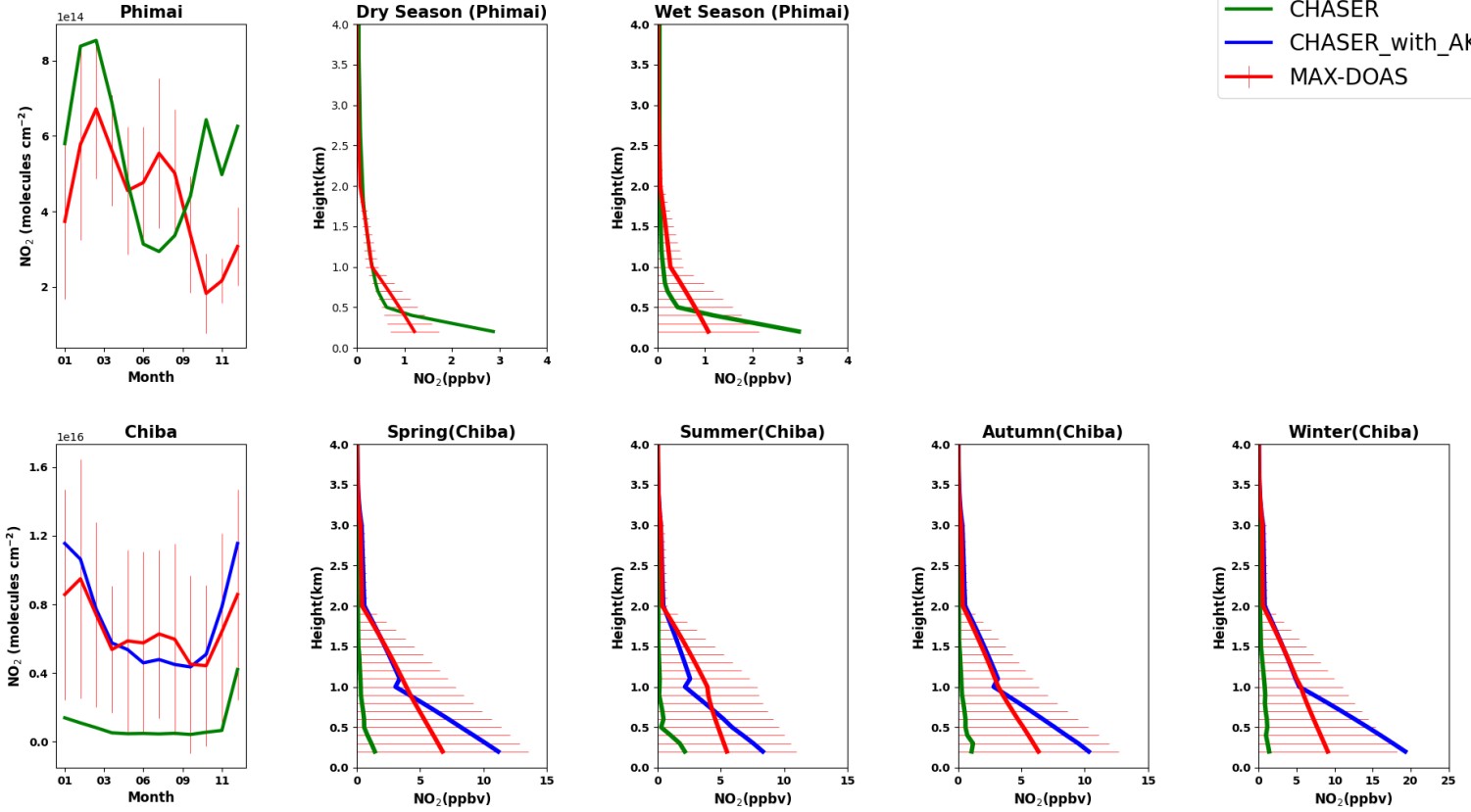


**Figure 11.** Seasonal variation in $NO_2$ partial columns from 0 - 4 km and vertical profiles during all seasons at Phimai and Chiba, as inferred from the MAX-DOAS observations (red) and CHASER simulation(green). The CHASER $NO_2$ partial column and vertical profile smoothed with the MAX-DOAS AK are coloured in blue. The coincident time and date between the model and observations are selected only. The error bars represent the one sigma standard deviation of mean values of the MAX-DOAS observations.






Table 7: Comparison of the seasonal mean $NO_2$ partial columns and profiles (0-4 km) between MAX-DOAS and CHASER at Phimai and Chiba. MBE (CHASER – MAX-DOAS) is the mean bias error. The partial column and profile MBE units are $\times 10^{15}$ molecules $cm^{-2}$ and ppbv, respectively.

| Site | Season | Partial column MBE | Smoothed Partial column MBE | Profile MBE | Smoothed Profile MBE |
|---|---|---|---|---|---|
| Phimai | Overall | 0.08 | | 0.11 | |
| Phimai | Dry | 0.18 | | 0.09 | |
| Phimai | Wet | -0.14 | | 0.02 | |
| Chiba | Overall | -5.58 | -1.90 | -3.27 | -1.66 |
| Chiba | Spring | -5.56 | -2.00 | -3.19 | -1.74 |
| Chiba | Summer | -5.52 | -2.87 | -2.85 | -1.86 |
| Chiba | Autumn | -4.57 | -1.24 | -2.74 | -1.40 |
| Chiba | Winter | -6.64 | -1.50 | -4.30 | -1.63 |

Although the datasets are moderately correlated ($R$=0.59) at Chiba, the model largely underestimates the $NO_2$ partial column with MBE of ~$5\times10^{15}$ molecules $cm^{-2}$. The model predicts almost constant $NO_2$ profiles and columns throughout the year. Therefore, the respective seasonal biases are almost similar. The vertical gradient of the modeled $NO_2$ profiles is also low, too, similarly to the HCHO profiles. The model resolution can be a potential cause for such significant underestimation. The AKs improved the partial column and profiles significantly, reducing the MBE by more than 50%. However, the smoothed profiles and partial columns between the 0 - 2 km layer, differ significantly from the simulations, suggesting that the a priori values strongly affect the smoothed profiles. Consequently, the smoothed $NO_2$ profiles at Chiba (Fig.S7) are biased to the a priori values, similar to that of Phimai (Fig. S6). $NO_2$ smoothed profile sensitivity to a priori values might be attributable to our retrieval procedure. The a priori data are taken from the measured SCD and retrieved VCD values. As a result, the values are sensitive in the 0 - 2 km layer, similarly to the observations. Using a priori values other than those obtained from observations can affect such sensitivity. The smoothing sensitivity to a priori values is stronger for $NO_2$ than HCHO. The $NO_2$ profile gradient is higher than that of HCHO (Figs. 10 and 11), which means that,

within 10 km (MAX-DOAS horizontal resolution), the $NO_2$ mixing ratio and a priori variability (sources and sinks) is higher than those of HCHO, leading to a stronger a priori effect on the smoothed profiles.

The mean $NO_2$ mixing ratios in the 0 - 2 km layer in 2018, simulated at spatial resolutions of 2.8° × 2.8° (standard) and 1.4° × 1.4°, are compared with observations at Chiba, as depicted in Fig.12. The error bars are the 1σ standard deviation of the observations. Higher resolution simulations reduced the overall MBE by 35% (Table 8). $NO_2$ concentrations at 1.4° are now within the variation range of the observations. The 1.4° simulation captured the $NO_2$ seasonal variability better than at 2.8°. Despite improved resolution, the model values are underestimated, with the highest MBE during the winter. According to Miyazaki et al. (2020), the seasonality in the anthropogenic emissions, primarily wintertime heating, is not well represented in the emission inventories, which could likely underestimate winter $NO_2$ levels. The best agreement between the datasets is observed during summer and spring, with an MBE of ~1 ppbv on a seasonal scale.

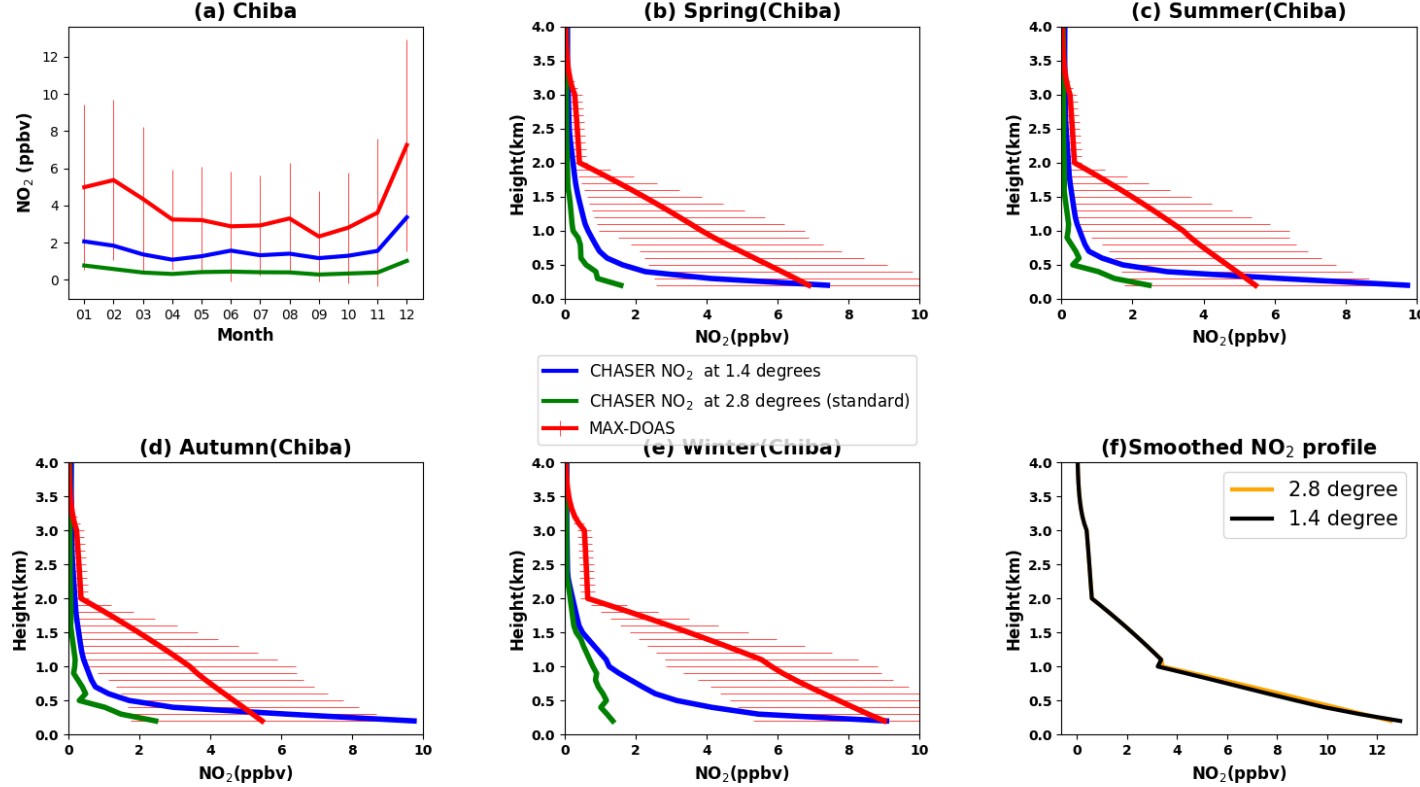

**Figure 12**: (a) Seasonal variations in the NO$_2$ mixing ratios in the 0 - 2 km layer at Chiba, as inferred from the MAX-DOAS observations (red) and two CHASER simulations at 2.8˚(green) and 1.4˚(blue) resolutions. The simulated NO$_2$ profiles at 2.8˚(green) and 1.4˚(blue) resolutions during (b) spring, (c) summer, (d) autumn, and (e) winter are shown with the observed seasonal profiles at Chiba. Only data (both observed and simulated) for 2018 are plotted. The coincident time and date between the model and observations are selected only. The error bars in (a), (b), (c), and (d) represent the one sigma standard deviation of mean values of the MAX-DOAS observations.

NO$_2$ profiles at 2.8˚ and 1.4˚ resolution are shown in Figs. 12(b - e). A strong effect of the increased resolution is observed below 500 m, reducing the negative bias by 70% near the surface. Above 500 m, the effects of higher resolution are limited, with an MBE reduction of 12% in the 0.6 – 2 km. Although the near-surface NO$_2$ concentrations at 1.4˚ resolution are overestimated, the values are within the standard deviation of the observations. At around 200m, winter mean NO$_2$ concentrations at 1.4˚

resolution are identical to the observations (~9 ppbv), and the summer mean is overestimated. Moreover, the $NO_2$ levels above 2 km are similar at both resolutions. The resolution effects on $NO_2$ profiles vary with the location and season (Williams et al., 2017). For example, CHASER $NO_2$ at 1.1° resolution improved the agreement with aircraft observations below 650 hPa significantly over the Denver metropolitan area (Sekiya et al. 2018), whereas, at Chiba, the 1.4° resolution improved the surface estimates. Consequently, the horizontal resolution is not the only reason for the model underestimation. Other factors such as the vertical resolution, uncertainties in emission inventories, and chemical kinetics, can also affect the simulated $NO_2$ estimates. Effects of the emission inventory is discussed in section 3.3.4.

Figure 12(f) shows the smoothed $NO_2$ profiles at both resolutions. Although the profile shapes are different, the smoothed profiles are almost identical, which demonstrates that, smoothed $NO_2$ profile sensitivity to a priori choice is mostly independent of the model resolution.

**Table 8:** Comparison of the seasonal mean $NO_2$ profiles (0-2 km) among MAX-DOAS and CHASER simulations at 2.8° and 1.4° resolutions at Chiba. MBE at (CHASER – MAX-DOAS) 1.4° and 2.8° are the mean bias error at the respective resolutions. The MBE unit is ppbv.

| Season | MBE at 1.4° | MBE at 2.8° |
|---|---|---|
| **Overall** | -2.24 | -3.37 |
| Spring | -2.26 | -3.23 |
| Summer | -1.50 | -2.47 |
| Autumn | -1.57 | -2.57 |
| Winter | -3.44 | -5.07 |

### 3.3.3 Evaluation of CHASER HCHO in the IGP region

The IGP is the most fertile region in South Asia, which accounts for approximately 50% of the total agricultural production of India and is one of the significant contributing regions to the global greening based on leaf area index (Sarmah et al., 2021). Moreover, IGP is one of the regional HCHO hotspots in India (Chutia et al., 2019). The observed HCHO seasonality at Pantnagar is consistent with that reported by Mahajan et al. (2015) for the entire IGP region. Consequently, comparison with the HCHO retrievals in Pantnagar can assess the model capability in the IGP region. The spatial representativeness is a limitation for comparison between a point measurement and regional simulations. Thus, the results are interpreted qualitatively. Because of the availability of a dataset with continuous observations, only the comparison for 2017 is shown in Fig. 13.

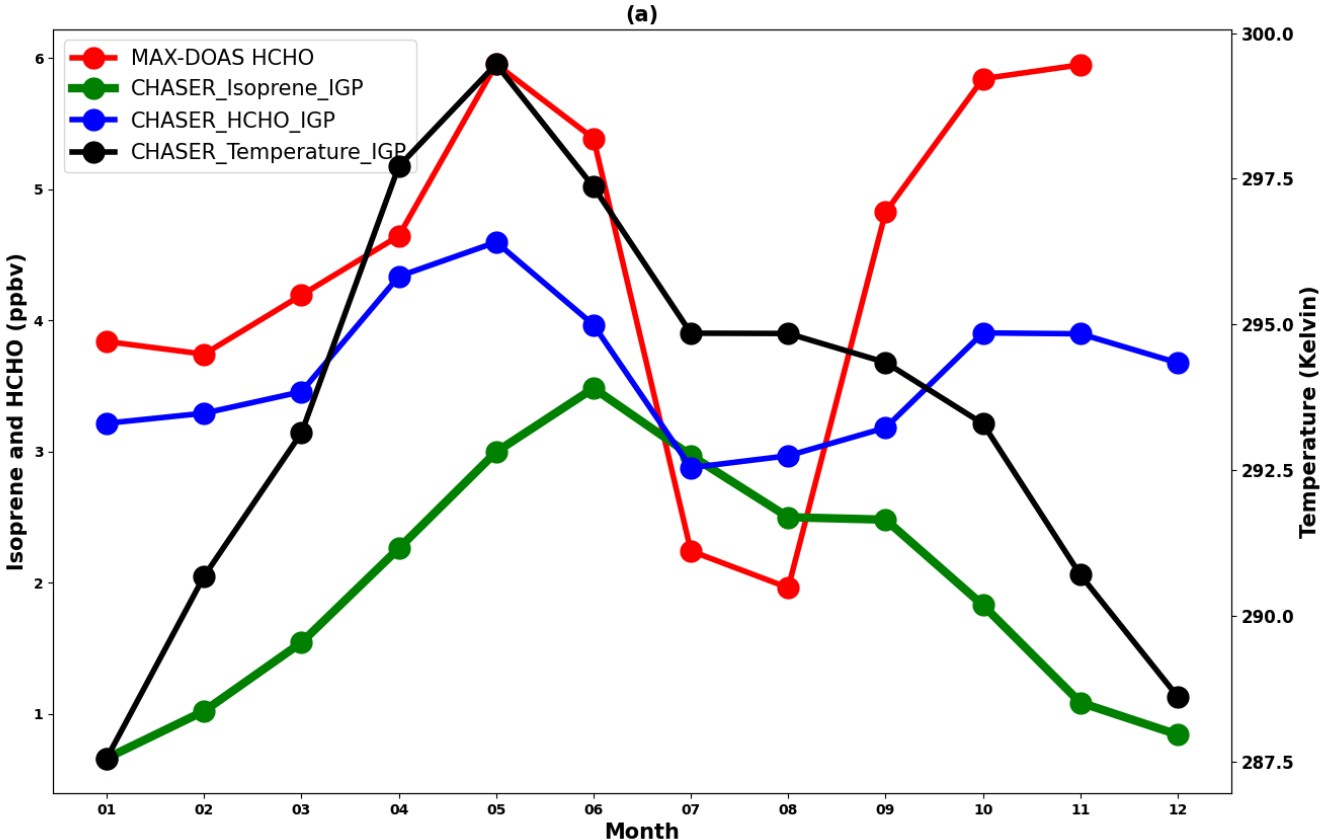

**Figure 13.** Seasonal variations in the MAX-DOAS (red) and CHASER (blue) HCHO concentrations at Pantnagar and the IGP region, respectively, in 2017. The coincident dates between the observations and model are plotted

only. The CHASER simulated isoprene and temperature seasonality are shown respectively, in green and black colours. Only the daytime simulated values were considered for the plot.

The modeled HCHO seasonal variations in the IGP region correlate well with the observations at Pantnagar ($R \sim 0.80$). The enhancement in the HCHO concentrations during the spring and post-monsoon season is well reproduced by CHASER, which indicates that CHASER can capture HCHO variation in complex terrain region such as IGP. Figure 13 also depicts the isoprene concentrations and temperature in the IGP region, in addition to the HCHO concentrations. Oxidization of precursor hydrocarbons and photochemical reactions are the most dominant sources of HCHO. Also, isoprene is the most abundant hydrocarbon in the atmosphere. The average ambient isoprene concentrations during July, August, and September in the IGP region are $1.4 \pm 0.3$ ppbv (Mishra et al., 2020). Therefore, the CHASER isoprene concentration range of $1.5 - 2$ ppbv during the monsoon season seems reasonable. The HCHO concentrations in the IGP region reach a peak during the spring and post-monsoon seasons. A strong correlation between HCHO, isoprene, and temperature variation ($R \sim 0.90$) during the first half of the year indicates that the change in biogenic emissions strongly drives the HCHO seasonal modulation. The observed enhancement in the HCHO levels during spring at Pantnagar is related to biomass burning. The biomass burning events are primarily concentrated in the northwest IGP region (Kumar and Sinha, 2021), where the site is located. On a regional scale, the biomass burning effects is expected to smear. Thus, the strong effect of the biogenic emission on the regional HCHO modulation is reasonable. HCHO modulation differs from isoprene and temperature during the post-monsoon period, suggesting a greater role of biomass burning and anthropogenic emissions. Consequently, the physical processes driving the HCHO seasonality in the IGP region are well reflected in the CHASER simulations.

### 3.3.4 Effects of the model resolution and emission inventories on results

Effects of the spatial resolution on the evaluation results is assessed by comparing the results of CHASER simulations at 2.8° and 1.4° resolutions with the surface observations, as shown in Fig. 14. Only, the simulated surface HCHO and $NO_2$ concentrations during 2017 are shown only. The statistics are provided in Table 9. For the Pantnagar site, only the simulations are presented. At Phimai, the HCHO simulations

differ by 3%. The standard simulation shows better agreement with the observations. The higher MBE at
1.4˚ occurred mostly because of the model overestimation during the wet season. The $NO_2$ mixing ratios
at the two resolution differ by 9%. The MBEs for both trace gases at Phimai are less than 1 ppbv. Thus,
the HCHO and $NO_2$ standard simulations at 2.8˚ can be regarded as reasonable for regions characterized
by low $NO_2$ levels (<1 ppbv). At Chiba, surface $NO_2$ and HCHO mixing ratios at 1.4˚ resolution differ
respectively, by 61 and 19%. The $NO_2$ MBE at 1.4˚ resolution improved significantly, indicating a strong
effect of the model resolution. However, discussion in section 3.2.2 showed limited resolution-based
improvement in the overall profile. Results for MBE in the HCHO mixing ratios at 1.4˚ mostly improved
during summer. The wintertime HCHO estimates at both resolutions are similar. In contrast to Chiba and
Phimai, differences in the HCHO simulations (30%) at Pantnagar are greater than those of $NO_2$ (3%).
The effect of model resolution varying with location and season was also reported by Sekiya et al. (2018).
Compared to the other two sites, differences in the $NO_2$ simulations at Chiba are larger. This finding is
consistent with the results by William et al (2017), which found larger differences with changing model
resolution over urban areas.
Although the $NO_x$ estimates for the low $NO_2$ regions seem reasonable, global $NO_x$ emissions have
changed since 2008(i.e., EDGAR-HTAP (2008) emissions used for this study). A recent study by Miyazaki
et al. (2020) reported changes in global $NO_x$ emissions from 2005 to 2018. They found a continuous 30%
increase in $NO_x$ emissions in India since 2005. REAS v3 (Regional Emission inventory in Asia version
3) inventory estimated a 23% increase in $NO_x$ emissions in India between 2010 - 2015, and power plants
were the most significant contributor. Many power plants are clustered along the IGP region (Nair et al.,
2007). Thus, the current simulation settings are likely to underestimate the $NO_2$ mixing ratios and columns
in the IGP region. Figure S8 presents comparison of CHASER and OMI $NO_2$ columns for 2017 over the
IGP region. Although the modeled columns are biased by 32% in the lower side, the spatial correlation
between the datasets is high ($r$=0.78). CHASER values lie within the range of variation of the
observations. Although underestimated, $NO_2$ estimates in the IGP based on the current inventory are yet
reasonable. Sekiya et al., (2018) used higher model resolution and updated emission inventory (HTAP
2010 for simulations in 2014) and reported ~30% lower MBE over India. However, the RMSE values of
both studies are comparable.
$NO_x$ emissions in Japan have shown continuous decline since the execution of pollution control policies
in 1970 (Ohara et al., 2020). Irie et al. (2021) reported a declining trend in $NO_2$ levels in Chiba since
2012, echoing results obtained by Miyazaki et al. (2020) throughout Japan. The bias between CHASER
and OMI $NO_2$ column over Japan is non-significant (Fig. S8 and Table S3). Thus, an updated inventory
will not substantially affect the comparison results at the Chiba site. $NO_x$ emissions increased
considerably in Southeast Asia. CHASER $NO_2$ estimates for Thailand based on HTAP 2008 inventory
are biased by 45% in the lower side compared to OMI (Fig. S8). However, Phimai being a rural site, the
$NO_x$ levels are expected to be low. Changes in biomass burning $NO_x$ estimates are likely to affect the
model estimates. Because, the $NO_2$ levels at Phimai are mostly less than 1 ppbv, the effect of updated
inventory on the comparison results is expected to be minimal.
CHASER HCHO columns over Japan, the IGP region, and Thailand are negatively biased respectively,
by 60, 36, and 32% compared to OMI observations, with $r$-values of 0.5 – 0.7 (Fig. S8). Surl et al. (2018)
reported spatial correlation of ~0.5 between GEOS-CHEM and OMI over the IGP region. Anthropogenic
VOC emissions in India and other Asian cities have increased since 2005, whereas a negative trend has
been observed over Japan (Bauwens et al., 2022). The REAS inventory estimated a 5% increase in
NMVOCS in India since 2005. Moreover, anthropogenic emission contributes strongly to the HCHO
abundances in the IGP region (Kumar and Sinha 2021). Thus, updated anthropogenic VOC emission
inventory is likely to improve the model HCHO estimates in the study regions. However, the formation
pathway of HCHO from isoprene emissions is a non-linear function of $NO_x$ chemistry. Consequently, the
effects of $NO_x$ emissions changes on the overall HCHO simulations cannot be assessed based on current
analyses explained herein.

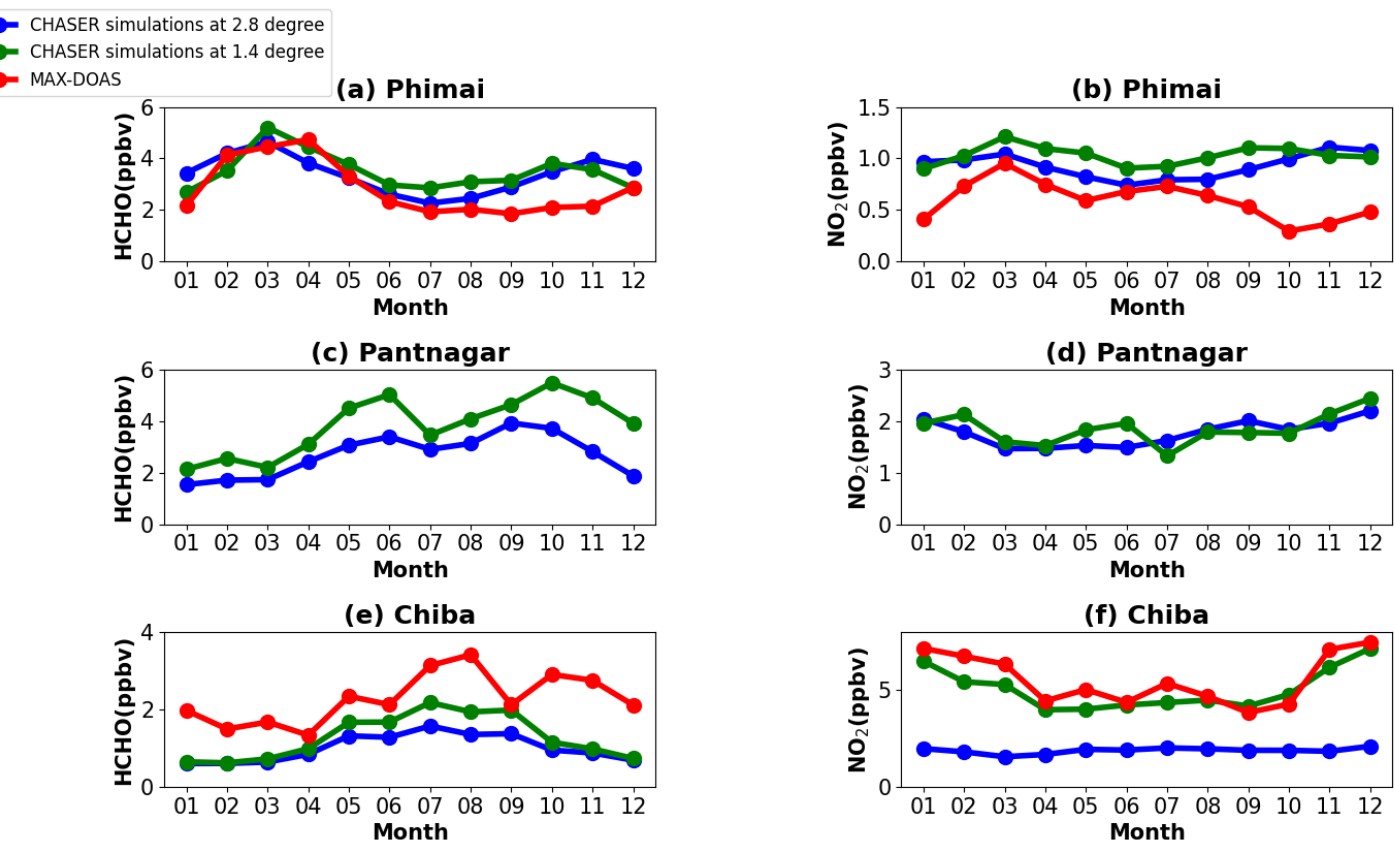

**Figure 14.** Seasonal variation in the surface HCHO and $NO_2$ mixing ratios at (a & b) Phimai, (c & d) Pantnagar, and (e & f) Chiba simulated at spatial resolutions of 2.8° ×2.8° (blue) and 1.4° × 1.4°(green). Coincident MAX-DOAS $NO_2$ and HCHO VMRs in the 0-1 km layer at Phimai and Chiba are plotted in red. Observation at Pantnagar are discarded. Only the datasets for 2017 are plotted.





**Table 9:** The comparison between the observations and simulations at 2.8˚ and 1.4˚ spatial resolutions. The MBE is the mean bias error. The unit of MBE is ppbv.

| Site | Trace gas | MBE at 2.8˚ | MBE at 1.4˚ | Differences between the simulations |
|------|-----------|-------------|-------------|-------------------------------------|
| **Phimai** | HCHO | 0.54 | 0. 65 | 3% |
| **Phimai** | NO$_2$ | 0.33 | 0.43 | 9% |
| **Chiba** | HCHO | -1.27 | -1.00 | 19% |
| **Chiba** | NO$_2$ | -0.52 | -3.69 | 61% |
| **Pantnagar** | | | | 30% |
| **Pantnagar** | | | | 3% |





### 3.4 Contribution estimates

### 3.4.1 Contribution from biomass burning to the HCHO and NO$_2$ abundances at Phimai

Good agreement between the datasets in the 0 - 1 km layer at Phimai can quantify biomass burning contributions to the HCHO and NO$_2$ concentrations. Figure 15 presents results of simulations L1_HCHO, L1_opt, and L1_NO$_2$. The simulation settings are presented in Table 3. For better readability, the switched-off emissions criterion is described in the legends of Fig.15. The plots present mean mixing ratios in the 0 – 1 km layer. Biomass burning contributes ~10% to the HCHO concentrations at Phimai during the dry season. However, based on the observations, a greater effect of biomass burning is expected. During the wet season, the MAX-DOAS and CHASER HCHO surface mixing ratios are, respectively, ~2 and ~4 ppbv (Fig. 10), indicating overestimation of the biogenic emissions in CHASER. Figure 15(b) shows the HCHO concentration obtained from simulation L1_opt and MAX-DOAS

observations in 2017. In the L1_opt simulation setting, the biomass burning emissions are switched off;
the biogenic emissions are optimized to reproduce results analogous to those obtained from observations
during the wet season. In the absence of biomass burning, the surface HCHO concentrations at Phimai
would be ~2 ppbv, indicating a biomass burning contribution of ~20–50% during the dry season. The
observed interseason difference in the HCHO concentration at Phimai is ~60%. Consequently, the revised
biomass burning contribution estimate is more reasonable. Pyrogenic emissions contributions to the $NO_2$
concentrations at Phimai are ~10% during the dry season (Fig. 15(c)). Because the $NO_2$ concentrations
are low at Phimai, the simulation results obtained for March, when the influence of biomass burning is
highest, are used to derive a better contribution estimate. In the absence of biomass burning, the $NO_2$
concentration during March would be about 0.84 ppbv (Fig.15(d)), indicating a contribution as high as
35% to the $NO_2$ concentrations at Phimai.

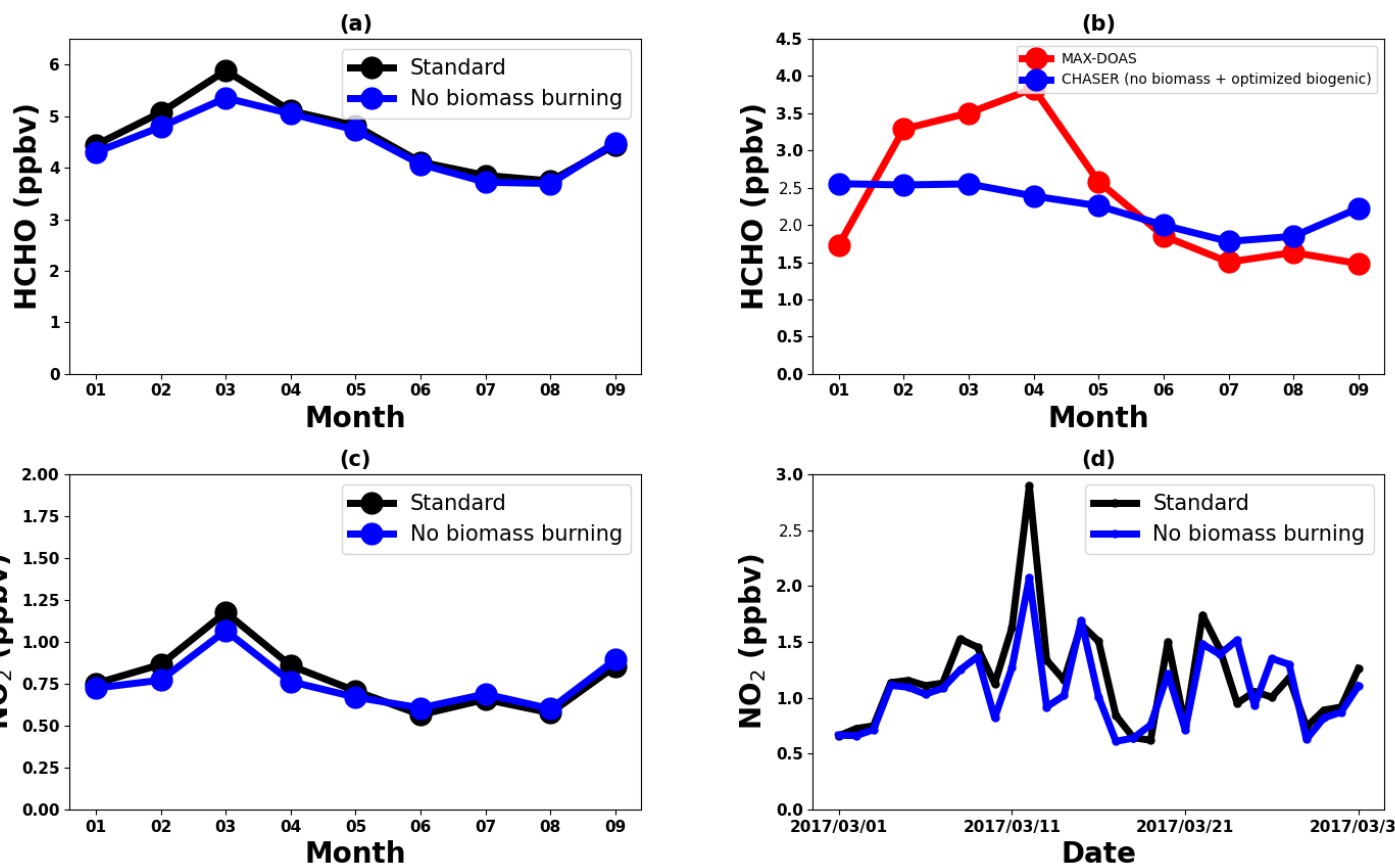


**Figure 15.** (top panel) (a) Seasonal variations in the HCHO concentrations in the 0 - 1 km layer at Phimai, as obtained from the standard and L1_HCHO simulations. Pyrogenic emissions of VOCs are switched off in L1_HCHO. (b) The HCHO seasonal variation in Phimai in 2017, as obtained from the MAX-DOAS observations (red) and L1_opt simulations. The pyrogenic VOC emissions were switched off, and the biogenic emissions were reduced by 50% in L1_opt. The coincident dates between the observation and the simulations are shown only. (bottom panel) (c) Seasonal variations in the $NO_2$ surface concentrations at Phimai in 2017, as obtained from the standard and L1_NO_2 simulations. (d) Standard and L1_NO_2 simulation outputs of the daily mean $NO_2$ surface concentrations during March 2017. The pyrogenic $NO_2$ emissions were switched off in the L1_NO_2 simulation. Only the daytime values from 09:00 – 15:00 LT are used to calculate the seasonal mean.

### 3.4.2 Contribution of soil $NO_x$ emissions at Phimai

Because soil $NO_x$ emissions are included in CHASER simulations, the $NO_2$ contributions from soil emissions are quantified. Figure 16 presents the monthly mean surface $NO_2$ concentrations at Phimai in 2017, simulated including (standard) and switching off (L1_NO_2) the soil $NO_x$ emissions. The $NO_2$ concentrations between 09 and 12 hr. were used to calculate the monthly mean concentrations. Soil emissions contribute ~20% of the overall $NO_2$ concentrations at Phimai, with higher contributions during the wet season. The highest soil contribution of about 25% occurs in July.

### 3.4.3 Contribution from pyrogenic and anthropogenic emissions to the HCHO abundances in the IGP region

Figure 16(b) presents the standard, L1_HCHO (pyrogenic VOC emissions switched off), and L2 (anthropogenic VOC emissions switched off) HCHO simulations in the IGP region. According to L1_HCHO simulation results, effects of biomass burning emissions on the regional HCHO modulation are small (~12%). The HCHO concentrations in India have biogenic, anthropogenic, and pyrogenic VOC sources. However, biogenic VOCs are the primary driver of the over HCHO variation (Surl et al., 2018). Consequently, two reasons might be responsible for the small effects of pyrogenic emissions on HCHO concentrations: (1) Overestimation of the biogenic emission or underestimation of pyrogenic emissions

in the model. (2) Stronger effects of anthropogenic VOC emissions than of pyrogenic VOCs. The L2 simulations show that anthropogenic emissions contribute up to 30% of the HCHO concentration in the IGP region, with a maximum contributed during the post-monsoon season, which coincides with the lower isoprene concentration (i.e., biogenic emissions) and temperature (Fig. 14). Moreover, Kumar and Sinha (2021) reported high acetaldehyde concentrations from anthropogenic emissions in the IGP region throughout the year. Consequently, anthropogenic emissions are likely to be a significant driver of HCHO concentrations in the IGP region after biogenic emissions.

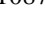

**Figure 16.** (a)Monthly mean $NO_2$ concentrations at Phimai were estimated from the standard (black) and L1_$NO_2$ (red) simulations. The soil $NO_x$ emissions are switched off in the LI_$NO_2$ simulation. The green line represents the percentage difference between the two simulations. (b) Seasonal variations in the HCHO concentrations in the IGP

region, obtained from the standard, L1_HCHO (pyrogenic VOC emission switched off), and L2 simulations (anthropogenic VOC emissions switched off). The simulations for 2017 are shown and analysed. Daytime values from 09:00-12:00 and 09:00 – 15:00 LT were selected respectively, for Phimai and IGP.

# 4 Conclusions

Using the JM2 algorithm, $NO_2$ and HCHO concentrations and profiles were retrieved from MAX-DOAS observations at three A-SKY sites during January 2017 - December 2018. The retrieved products were used to evaluate the global chemistry transport model CHASER simulations at the three sites. At all three locations, the seasonal variation of both trace gases was consistent throughout the investigated period. At Phimai and Pantnagar, biomass burning led to enhanced HCHO and $NO_2$ concentrations, respectively, during the dry season and spring and post-monsoon season. At Chiba, the HCHO variation was consistent with the temperature-led seasonal changes in biogenic emissions. The changes in the dry season HCHO and $NO_2$ levels at Phimai during 2015 - 2018 were consistent with the number of fire events.

The $R_{FN}$ values were biased towards a particular regime when the standard transition range $1 < R_{FN} < 2$ (Duncan et al., 2010) was used. The parameterization of Souri et al. (2020) provides a better estimate of the transition region. The classification results of the revised transition region at Phimai and Pantnagar contradicted the results based on the standard transition range. However, they were more reasonable. Such a method based on observations, is therefore influenced by measurement constraints. More observational evidence must be accumulated to standardize this method. Overall, the results further indicated that that the standard transition region is not valid globally.

Despite the use of an old $NO_x$ emission inventory the simulated $NO_2$ and HCHO spatial distributions agreed reasonably well with those observed from satellite- observations. The modeled regional $NO_2$ columns estimates were within the 2-sigma variability range of OMI $NO_2$ retrievals. Although the negative bias in HCHO comparison was higher than that of $NO_2$, the model demonstrated good capabilities for simulating the HCHO seasonal variation in different regions.

CHASER showed good capabilities at Phimai, characterized as a VOC-rich and low $NO_2$ (<1 ppbv) region. In both seasons, the observed and modeled profiles (HCHO and $NO_2$) agreed within the one sigma

standard deviation of the measurements, despite general overestimation of the model. Furthermore, both wet season HCHO profiles were almost identical in the 0.5 – 4 km layer in both datasets. CHASER demonstrated limited performances at Chiba. $NO_2$ at higher resolution (i.e.,1.4˚) mainly improved the surface estimates, reducing the overall MBE in the 0 - 2 km layer by 35%. Finer resolution would improve the HCHO estimates in Chiba by 10%; however, it has yet to be underestimated. Sensitivity studies for the Phimai site estimated biomass burning contributions to the respective HCHO and $NO_2$ concentrations up to ~50 and ~ 35%, respectively. On average, 20% of the $NO_2$ level originates from soil $NO_x$ emissions, which increased to 25% in July. Anthropogenic emissions (contribution up to 30%) have a more strongly affect VOC variation in the IGP region than biomass burning, which is consistent with reports presented in the literature.

*Code availability.* The CHASER and JM2 source codes are not available publicly. Dr. Kengo Sudo (kengo@nagoya-u.jp) is the contact person for readers and researchers interested in the CHASER model. In addition, Dr. Hitoshi Irie (hitoshi.irie@chiba-u.jp) will answer queries related to the JM2 codes.

*Data availability*: The MAX-DOAS data used in the study are publicly accessible on the A-SKY network website (http://atmos3.cr.chiba-u.jp/a-sky/data.html). Upon request, the corresponding author can provide the CHASER simulations and MAX-DOAS averaging kernel data.

*Author contributions:* HMSH conceptualized the study, conducted the model simulations, analysed the observational and simulation data, and drafted the manuscript. AMF helped with the data processing. HI developed the JM2 code and maintained the A-SKY network. KS developed the CHASER model and supervised the study. MN is the PI of the Pantnagar site. AD and MN shared their experience to explain the results. HI, KS, AD, MN, and AMF commented and provided feedback on the final results and manuscript.

*Conflict of Interest*: The authors declare that they have no conflict of Interest

*Acknowledgments*: This research is supported by the Global Environmental Research fund (S-12 and S-20) of the Ministry of the Environment (MOE), Japan, and JSPS KAKENHI Grants: JP20H04320, JP19HO5669, and JP19H04235. The CHASER model simulations are partly performed with the supercomputer (NEC SX-Aurora TSUBASA) at the National Institute for environmental studies (NIES), Tsukuba, Japan. The authors are grateful to the OMI and TROPOMI data providers. Support from ISRO-ATCTM project for Pantnagar site is also acknowledged.

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
