# Peer review of "MAX-DOAS observations of formaldehyde and nitrogen dioxide at three sites in Asia and comparison with the global chemistry transport model CHASER"

_Atmospheric Chemistry and Physics, 2022_

## Author Comment (AC1)

**Responses to the comments of Reviewer 1**

We thank the referee for the helpful comments. We have revised the manuscript accordingly. The responses (blue fonts) are provided after stating the reviewer comments. Figure, Table, and line numbers correspond to the revised manuscript. The highlighted text are corresponding changes in the revised manuscript.

In this paper, formaldehyde (HCHO) and nitrogen dioxide (NO2) vertical profiles are retrieved from MAX-DOAS observations at three sites in Asia from January 2017 through December 2018. The three sites are Phimai in Thailand, Pantnagar in India, and Chiba in Japan. They correspond to rural, semi-urban, and urban conditions, respectively. The NO2 and HCHO concentrations in the 0-4km altitude range show consistent seasonal variations throughout the investigated period, which are interpreted in terms of dry and wet seasons and, in the case of Phimai and Pantnagar, biomass burning episodes. The HCHO to NO2 concentration ratios together with MAX-DOAS ozone retrieval results are also used to infer the ozone sensitivity to NOx and VOCs at the three sites. It is found that reasonable estimates of transition regions between the NOX-limited and VOC-limited ozone production regimes can be derived when the NO2-HCHO chemical feedback is accounted for.

In the second part of the study, the MAX-DOAS observations of NO2 and HCHO are used to assess the CHASER global CTM at the three sites. CHASER shows reasonably good performances in reproducing the abundances of both trace gases in Phimai and Pantnagar but not in Chiba. Comparison results are interpreted in terms of model resolution, emission inventories, and contributions of the different emission sources.

This study fits with the scope of ACP. However, there are a lot of aspects of the work that should be further clarified and/or discussed prior to final publication. Those aspects are detailed below. Moreover, as already raised during the quick review, the overall presentation quality is questionable, largely due to the poor English language used throughout the manuscript but also to repeated errors in the axes and title labeling of several figures. This should be improved in the revised version of the paper.

**Response:** We thank the reviewer for the comments which helped to improve the quality of the manuscript. In addition to the specific comments, the revised manuscript has been proof-read by a professional proof reader. Moreover, the figures have been improved according to the reviewer comments

**Important specific comments:**

*Lines 225-235: The VCD retrieval is based on several assumptions that are poorly discussed and justified. For instance, did you check that the dependence of the Abox profiles on the trace gas concentration profiles is indeed minimal? Did you test other a priori VCD values? How valid is assuming an Angstrom exponent value of 1.00?

**Response:** We thank the reviewer for the comments. Firstly, yes, sensitivity test assuming 30 and 50% uncertainties in the $A_{box}$ profiles were performed and no significant changes in the results were observed. Such values (i.e., 30 and 50%) were estimated empirically from comparison with sky radiometer and LIDAR observations. Rather than the word "minimal" we use the word "low" in the revised manuscript. Because, we can't judge whether the dependence is minimal, despite optimal $A_{box}$ profiles. Secondly, yes, we tested different a priori VCD values. Because the area in the averaging kernel was close to 1, the retrieval was almost independent of the a priori values. Thirdly, the choice of the Angstrom values had non- substantial impact on the retrieval. Uncertainty related to the Angstrom value was smaller than the uncertainties in the $A_{box}$ profiles. The following texts are added in the revised manuscript.

L 254 – 257: The choice of the Angstrom exponent value can induce uncertainty in the retrieved VCDs. However, such uncertainty was found to be non-significant compared to that of $A_{box}$ profiles. Uncertainty in the $A_{box}$ profiles are assumed to as high as 30 to 50%. Such values are derived empirically from comparison with sky radiometer and LIDAR observations (i.e., Irie et al., 2008b).

*Lines 234-236: You should describe how these averaging kernels are calculated. Looking at Figure 3, HCHO and NO2 VCD averaging kernels seem to be close to unity but it is not the case for f1 averaging kernels, and especially f2 and f3 averaging kernels which are close zero. Does it mean that you can basically retrieve only the VCDs from your measurements and that for f1, f2, and f3 the retrieval essentially reproduces the a priori? Also, are similar averaging kernels obtained for the two other stations? These points should be further discussed in the revised manuscript.

**Response:** We thank the reviewer for the comments. Firstly, the F values determine the profile shape. For example, an a priori F1 values of 0.6 means than 60% of the aerosol/trace gas is located below 1 km. If the F value is close to 1, it means the 100% of the aerosol/trace gas is located below the specified height. Thus, even at lower F values, a realistic profile can be derived in our retrieval. The impact of uncertainties in the F values has been discussed in Irie et al. 2008. Fig S2 (supplementary information) shows retrieved aerosol profiles with different F values. Secondly, averaging kernels for the other sites are shown in Fig in the supplementary information. The following changes were included in the manuscript.

L 263 – 270: Figure 3 presents the mean averaging kernel (AK) of the HCHO and $NO_2$ retrievals during the dry season at Phimai. The area (Rodgers, 2000) provides an estimate of the measurement contribution to the retrieval. The total area is the sum of all the elements in the AK and weighted by the a priori error (Irie et al. 2008a). The areas for VCD and *f1* of $NO_2$ retrieval are 1 and 0.6, respectively. The *f2* and *f3* values are much smaller. Consequently, at first, the a priori profiles were scaled, and later f values determined the profile shape. The VCD area is close to unity, and therefore, the retrieved VCD is

independent of the a priori values. Irie et al (2008) conducted sensitivity studies of choice of the *f* values and reported the effect on the retrieval negligible.

*Line 284: Anthropogenic emissions used in the CHASER model were based on the HTAP_v2.2 2008 inventory. Why didn't you use more recent inventories like the REAS v3 one (see https://acp.copernicus.org/articles/20/12761/2020/acp-20-12761-2020.pdf)? How can it affect the results and conclusions of your study, especially for Pantnagar and Chiba?

**Response:** We thank the reviewer for comments. We haven't yet updated the inventories due to some technical issues and currently we are trying to resolve the problem. An old inventory will have some impact on our results. We have revised the discussion on emission inventory impact in section 3.3.4. Moreover, we have provided evaluation of simulations against satellite observations in section 3.2.

*Figure 9: How do you explain such a large effect when model profiles are smoothed with the MAX-DOAS averaging kernels, especially in the altitude range (0-2km) where the MAX-DOAS retrievals have a maximum of sensitivity.

**Response:** We thank the reviewer for the comments. The large effect of the smoothing of the model profiles are likely related to the apriori values used for smoothing. Because the apriori data are the taken from the measured SCD and retrieved VCD values, it is sensitive in the 0-2 km layer, similar to the observations. Utilizing apriori values other than from observations will potentially impact such sensitivity. Figure 9 is Figure 11 in the revised manuscript. The following discussion has been included in the revisions:

L 846 -857 : The AKs improved the partial column and profiles significantly, reducing the MBE by more than 50%. However, the smoothed profiles and partial columns between the 0 - 2 km layer, differ significantly from the simulations, suggesting that the a priori values strongly affect the smoothed profiles. Consequently, the smoothed $NO_2$ profiles at Chiba (Fig. S5) are biased to the a priori values, similarly to that of at Phimai (Fig. S5). $NO_2$ smoothed profile sensitivity to a priori values might be attributable to our retrieval procedure. The a priori data are taken from the measured SCD and retrieved VCD values. As a result, the values are sensitive in the 0 - 2 km layer, similarly to the observations. Using a priori values other than those obtained from observations can affect such sensitivity. The smoothing sensitivity to a priori values is stronger for $NO_2$ than HCHO. The $NO_2$ profile gradient is higher than that of HCHO (Figs. 10 and 11), which means that, within 10 km (MAX-DOAS horizontal resolution), the $NO_2$ mixing ratio and a priori variability (sources and sinks) is higher than those of HCHO, leading to a stronger a priori effect on the smoothed profiles.

*Figure 10 and related discussion: What would be also interesting to show are model profiles at both 2.8°x2.8° and 1.4°x1.4° resolution smoothed by the MAX-DOAS AVK. I think only

this comparison allows to discuss quantitatively the effect of the model resolution on the CHASER/MAX-DOAS agreement. Since the 2.8°x2.8° and 1.4°x1.4° model profiles have a significantly different shape, we can expect a different impact when those profiles are smoothed with the AVKs.

**Response:** We thank the reviewer for the comments. An additional with smoothed $NO_2$ profiles at 1.4 resolution has been included in the revised manuscript. However, due to the strong impact of the apriori values, the differences are non-significant. Figure 10 is Figure 12 in the revised manuscript. The following discussion has been included.

L 892 – 894: Figure 12(f) shows the smoothed $NO_2$ profiles at both resolutions. Although the profile shapes are different, the smoothed profiles are almost identical, which demonstrates that, smoothed $NO_2$ profile sensitivity to a priori choice is mostly independent of the model resolution.

*Section 3.2.3: Given the coarse horizontal resolution of the CHASER model (2.8°x2.8°), how valid is the assessment of NO2 and HCHO from this model for the Pantnagar station which is located in a region (Himalayan foothills) with highly varying topography? I would suggest to remove Pantnagar from the model evaluation since the topography is not properly taken into account in your analysis.

**Response:** We thank the reviewer for the comments. We have removed Pantnagar from the model evaluation.

Minor comments:

*Line 125: You should indicate here which types of industries are located in the Pantnagar region.

**Response:** We have added the following texts:

L136 – 139 : Rudrapur (~12 km south-west of Pantnagar) and Haldwani (~ 25 km north-east of Pantnagar) are the two major cities near Pantnagar, where industries (Fast moving Consumable Goods, electroplating, plywood, pharmaceuticals, automobile and allied industries (Banerjee and Srivastava 2009)) are located.

*Lines 151-152: The use of the 70°EL instead of the 90°EL for the reference spectra should be better justified. How the use of 70°EL (instead of 90°EL) can minimize variations in the measured signals. Also what do you mean by 'variations in the measured signals'?

**Response:** Firstly, we used a spectrometer with a fixed integration time throughout the day. The intensity of the spectra usually depends on the elevation angle (EL) within a 15-min interval (time for a complete scan for all EL). Thus, the variation range of intensities measured at all ELs can be large, which occasionally leads to intensity saturation at the reference angle. To avoid such phenomenon, the refence measurement were conducted at 70˚ instead of 90˚. All the ELs were considered in the differential air mass factor calculation to retrieve the vertical profiles. Thus, the choice of reference EL (70˚ or 90˚) is not a critical issue. Secondly, " variation in the measured signals" has been replaced with " to avoid saturation of intensity". The changes/addition in the revised manuscript is as follows:

L163-167: The sequences of the ELs at all the sites were repeated every 15 min. The reference spectra are recorded at EL of 70° instead of 90° to avoid saturation of intensity. Because all the ELs were considered in the box air mass factor ($A_{box}$) calculation to retrieve the vertical profile, the choice of reference EL (70˚ or 90˚) is not an important issue for this study.

*Line 196: You should give here the AEC value at 100km you used, as well as the scaling height of your exponentially decreasing a priori profile.

**Response:** The following changes corresponding to this comment has been included in the revised manuscript.

L 211- 216: The AEC profile from 3 to 100 km is derived assuming a fixed value at 100 km and exponential AEC profile shape with a scaling height of ~1.6 km. The k value at 100 km was estimated from Stratospheric Aerosol and Gas Experiment III (SAGE III) aerosol data ($\lambda$=448 and 521 nm) taken at altitudes of 15–40 km. The non-substantial influence of such assumptions on the retrievals in the lower troposphere has been demonstrated in sensitivity studies reported by Irie et al (2012).

*Lines 201-203: The parameterization of Irie et al. (2008a) does not provide information on the vertical resolution and measurement sensitivity. Then it is said that 'The retrievals and simulations conducted by other groups for similar geometries (i.e., Frieß et al., 2006) are used to overcome such limitations. I don't understand this latter sentence. Do you mean that you used previous studies based on the optimal estimation method to estimate the vertical resolution and sensitivity of your own parameterized retrieval? Could you please clarify?

**Response:** We thank the reviewer for the comments. The limitation of our retrieval is that the vertical sensitivity can't be derived instantly. Thus, we estimate the vertical sensitivity from other studies using the similar geometry. Our retrievals using such an approach has been validated with other ground-based observations (Irie et al 2012; Irie et al. 2015). Moreover, multi-component retrievals adopting a similar approach has been reported in details by Irie et al. (2008a).

*Lines 206-208: You should describe in a table the settings (pressure and temperature profiles, wavelength, surface albedo, etc) you used for the calculation of your box air mass factors LUT.

**Response:** Instead of a table we have mentioned the parameters in the revised manuscript as follows:

L227-235 : Then, a lookup table (LUT) of the box air mass factor ($A_{box}$) vertical profile at 357 and 476 nm is constructed using the radiative transfer model JACOSPAR (Irie et al., 2015), which is based on the Monte Carlo Atmospheric Radiative Transfer Simulator (MCARaTS) (Iwabuchi, 2006). The values of the single- scattering albedo (s), asymmetry parameter (g), and surface albedo were, respectively, 0.95, 0.65 (under the Henyey-Greenstein approximation), and 0.10. The U.S. standard atmosphere temperature and pressure profiles were used for radiative transfer calculations. Uncertainty of less than 8% related to the usage of fixed values of s, g, and a were estimated from sensitivity studies (i.e., Irie et al 2012). Results obtained from JACOSPAR are validated in the study reported by Wagner et al. (2007). The optimal aerosol load and the $A_{box}$ profiles are derived using the $A_{box}$ LUT and the $O_4$ ΔSCD at all ELs.

*Lines 244-245:  For the estimation of the systematic errors, uncertainties of 30% and 50% on the retrieved AOD are assumed. Where these uncertainty values come from?

**Response:** The uncertainties of 30 and 50% has been derived empirically from comparison with sky radiometer and LIDAR observations (Irie et al., 2008).

*Lines 246-247: Did you try to estimate the presence of an EL bias e.g. by performing horizon scans on a regular basis?

**Response:** We thank the reviewer for the comments. The bias in the ELs were estimated from retrievals using additional $A_{box}$ calculations, assuming ±0.5° shifts in the ELs. The detail has been explained in Hoque et al (2018).

*Lines 252-254: The criteria used for the cloud screening should be justified. How do you determine them?

**Response:** The following text has been added corresponding to the comment.

L287-288: The threshold values were determined statistically corresponding to the mode plus one sigma (1σ) in the logarithmic histogram of relative residuals.

*Lines 289-290: Where these emission values come from? References or justification are needed here.

**Response**: Emission values are taken from the biogenic emission inventory VISIT used in the model. The sentence has been revised:

L326-327: Isoprene, terpene, acetone, and ONMV emissions estimates in the VISIT inventory during July were $2.14 \times 10^{-11}$, $4.43 \times 10^{-12}$, $1.60 \times 10^{-12}$, and $9.93 \times 10^{-13}$ $kgCm^{-2}s^{-1}$.

*Lines 397-398: In Figure 5, only the $O_3$ concentrations for SZA < 50° are used to minimize stratospheric effects. Does it mean that only HCHO and NO2 data corresponding to SZA lower than 50° have been selected for these plots? If not, this means that HCHO and $NO_2$ retrieval results does not timely coincide with the $O_3$ concentrations. This point should be clarified.

**Response:** We thank the reviewer for the comment. Yes, SZA < 50 criterion has been applied for all the three datasets.

*Line 398: It is stated that the JM2 O3 product showed good agreement with ozonesonde measurements. Has such verification been done at the three stations involved in the present study? Also, the Irie et al. (2021) reference is missing in the list.

**Response:** Firstly, appropriate ozonesonde measurements are not available for the sites used in the study. Because, the retrieval settings are similar for all the sites (including that of mentioned in Irie et al . (2021)), we expect a similar quality of the retrievals. Secondly, the missing reference has been included in the reference list.

*Figure 7(a): Even if they both correspond to high O3 concentration conditions, I am surprised to see that the Rfn vertical profiles at Phimai and Pantnagar have both the same shape. Could you comment on this point? Also, why the Rfn vertical profiles from the CHASER model are not included in Figures 7(a) and (b)?

**Response:** We thank the reviewer for the comments. The high $O_3$ concentrations in Phimai and Pantnagar occurs due to biomass burning, and thus a similar $R_{FN}$ profile is observed. The $R_{FN}$ profiles from the CHASER model are not included because – (1) we only focused on the $R_{FN}$ profiles obtained from the observations, and (2) to discuss the $R_{FN}$ profiles, the $O_3$ simulations should be included, which is out of the scope of the current work. However, such comparisons will be discussed in our future studies.

*Section 3.2.1: I think it would be useful to show the seasonally-averaged MAX-DOAS AVK corresponding to the climate classifications of each site in the Supplement. This would support the discussion here.

**Response:** We thank the reviewer for the comments. We have provided the seasonal averaged AVKs for the Phimai and Chiba site in the Supplementary information (Fig S4). The discussion on the Pantnagar site has been discarded, thus, the AVKs for Pantnagar are not included.

*Figure 8: given the very large error bars on the MAX-DOAS vertical profiles, I think it is important to say that the CHASER with AK – MAX-DOAS differences are not statistically significant.

Response: We have included the following sentence.

L797 – 798 :Overall, the differences between the observations and smoothed profile are statistically insignificant.

*Section 3.2.3: Why no CHASER versus MAX-DOAS profile comparisons are shown for NO2 and HCHO for Pantnagar? This is not consistent to what is presented at the Phimai and Chiba stations.

**Response:** The discussion on the Pantnagar site has been discarded due to the complex topography of the site.

*Line 744: Is it 1.1° or 1.4°?

**Response:** It is 1.1˚ according to Sekiya et al., (2018)

Technical corrections:

*Line 24: 'variation' -> 'variations'

**Response**: The word was corrected appropriately.

*Line 29: 'good performances reproducing' -> 'good performances in reproducing'

**Response**: The sentence was corrected appropriately.

*Line 48; 'the lifetime' -> 'the lifetime of HCHO'

**Response**: The phrase was corrected appropriately.

*Line 78: 'satellite retrieval' -> 'satellite data retrievals'

**Response**: The wording has been corrected

*Lines 97-98: 'in three atmospheric environments' -> 'in three different atmospheric environments'.

**Response**: The wording has been corrected

*Figure 1, page 6: I would use 'concentration' instead of 'concentrations' in the legend of the color bar.

**Response:** We have replaced concentration/concentrations to volume mixing ratio following reviewer 2's comments.

*Line 144: 'campaign' -> 'campaigns'

**Response**: The wording has been corrected

*Line 147: 'consist' -> 'consists'

**Response**: The wording has been corrected

*Line 164: 'following equation.' -> 'following equation:'

**Response**: The punctuation has been fixed

*Lines 174-175: 'cross section data' -> 'cross section data sources'

*Line 181: 'using the optimal estimation method (Irie et al., 2008a; Rogers, 2000)' -> 'using the approach developed by Irie at al. (2008a) which is based on the optimal estimation method (Rogers, 2000).'

**Response**: The sentence has been revised

*Line 182: 'In this approach, the measurement vector y….are defined as'

**Response**: The sentence has been revised

*Line 188: 'window' -> 'windows'

**Response**: The wording has been corrected

*Line 192: 'compromise' -> 'includes'

**Response**: The wording has been corrected

*Figures 5 and 6: It is not clear to me why the y-axis scales of the three plots are not the same in both figures. Please comment. Also, to my opinion, only the transition lines should change between figures 5 and 6, so one unique figure including the three transition lines should be fine.

**Response:** We have merged the figures which is figure 5 in the revised manuscript. A consistent y-axis scale has been used.

*Line 458: 'clarify' -> 'support'

**Response**: The wording has been corrected

*Page 554: 'imitate' -> 'reproduce'

**Response**: The wording has been corrected

*Figure 9: 'HCHO' should be changed to 'NO2' in the x-axis label of all plots.

**Response:** Fig.9 is Fig.11 in the revised manuscript and the axis—label has been corrected.

*Figure 10(b): I guess the blue and green curves should be inverted (green curve should be in blue and the blue curve in green).

**Response:** Figure 10(b) has been corrected and is Fig 12 in the revised manuscript.

*Figure 11: the same x-axis scale should be used in the four plots.

Response: The comparison discussion on the Pantnagar site has been discarded. Figure 11 is Fig. 13 in the revised manuscript.

*Line 822: 'Biogenic' -> 'biogenic'

**Response:** Appropriate corrections has been included in he revised manuscript

*Legend of Figure 14(b): 'no anthrpogenic' -> 'no anthropogenic

**Response:** The legend has been revised. Fig. 14 is Fig 16 in the revised manuscript.

---

## Author Comment (AC2)

**Responses to the comments of Reviewer 2**

We thank the referee for the helpful comments. We have revised the manuscript accordingly. The responses (blue fonts) are provided after stating the reviewer comments. Figure, Table, and line numbers correspond to the revised manuscript. The highlighted text are corresponding changes in the revised manuscript.

The manuscript by Hoque et al. shows MAX-DOAS measurements of $NO_2$ and HCHO at three sites in Asia, namely Phimai (Thailand), Pantnagar (India) and China (Japan). The MAX-DOAS measurements are compared with the global chemistry model CHASER simulated concentration in the near-surface layer as well as the profiles. An attempt was made to use the ratio of Formaldehyde and $NO_2$ concentrations to derive ozone production sensitivity.

While I have mentioned some critical concerns about the significance of this study with respect to the Journal in my short review before the discussion phase, I provide my elaborate review here. Most likely, the short review prior to the discussion phase is not available in the interactive discussion; I append that here and expect it to be addressed.

**Response:** We thank the reviewer for the comments. The comments during the short review has been addressed herein.

Broadly the paper covers two separate aspects, namely MAX-DOAS measurements and comparisons with the global model. On the one hand, there are some shortcomings in both aspects of this study; I also find it difficult to motivate the readers, why such a comparison should be made in 2022, and what do we expect to learn from it. For a comprehensive evaluation of the global model, a global dataset (e.g. NDACC) should be used, which are also recently employed to evaluate TROPOMI data products (e.g. (De Smedt et al., 2021; Lerot et al., 2021; Verhoelst et al., 2021) ). If the study is focused on south-east Asia, why a regional model with a better spatial resolution is not used? Several previous studies have used high resolution (few km), global models, for comparison with MAX-DOAS measurements and emphasised the need to even go for higher spatial resolution (sub km). This study, on the other hand, presents the model results at 2.8° resolution in the base case and 1.4° in the improved resolution case, which in my opinion, is too coarse for comparison with MAX-DOAS measurements.

**Response:** We thank the reviewer for the comments. Here are our responses:

(1) It is true that, similar studies have been reported in literature using different global and regional models. However, similar type of comparison is still important to evaluate many existing models. Because, simulation from different models differ depending on various model parameters and mechanism.

(2) Firstly, our research group focuses on various global scale study using the CHASER model. Research themes include global-scale chemistry, satellite comparison, and data assimilation. Thus, evaluation of regional/local simulations is important to assess model uncertainties and limitations, as conducted in earlier studies. Secondly, the choice of the regions is directly related to our MAX-DOAS observation facility, i.e., the A-SKY network. We agree with the reviewer that a regional model would be a better option. However, our research objectives are defined based on our existing modeling and observation facilities.

(3) We agree with the reviewer that; global models should be evaluated with global datasets. We have included the evaluation of global $NO_2$ and HCHO simulations with OMI observations in the revised manuscript.

(4) Earlier studies have indeed emphasized on higher model resolutions. High resolution models are preferable, however, not widely available/affordable due to technical limitation and expenses. Moreover, horizontal model resolution is not the sole the reason for discrepancies in comparison studies involving simulations. For example, the global mean bias (MB) in the CHASER $NO_2$ simulations at 2.8˚ and 0.56˚ compared to OMI were respectively, -0.25 and -0.24 x $10^{15}$ molecules $cm^{-2}$ (Sekiya et al 2018). Over SE-Asia the MB and root-mean square error at 2.8 and 0.56 were ~0.54 and 0.61 x $10^{15}$ molecules $cm^{-2}$ (Sekiya et al 2018), respectively. Moreover, model resolution impact on comparison result varies spatially (William et al 2017). Most of the global studies involving CHASER is conducted at 2.8 resolution, which is our standard model settings. Our objective is to evaluate the standard model simulations with the A-SKY observations. The impact of the model resolution on the results has been assessed in section 3.3.4.

Concerning the drawbacks related to MAX-DOAS retrievals, I find the vertical grid resolution (1km) too coarse, which limits the usability and interpretation of such data for air pollution-related studies. There are some technical issues related to the measurements as well, but those should be discussed in a detailed review if the editor deems the manuscript suitable for discussion in ACPD.

**Response:** We thank the reviewer for the comments. We have assumed an exponential profile and considering the fact that vertical resolution depends on the aerosol loading, we think our retrieval settings are reasonable. Using such settings, our aerosol and trace gas concentrations has shown good agreement with surface concentrations (Damiani et al 2021; Irie e al 2008, 2015,2012) and satellite observations (Irie et al 2012). Moreover, many air-pollutions related studies have been conducted using the similar settings (Irie et al 2016, 2019, 2021, Hoque et al 2018a, 2018b). Additional technical issues have been addressed in the detailed review responses.

 Damiani et al, (2021), Variabilities in PM2. 5 and Black Carbon Surface Concentrations Reproduced by Aerosol Optical Properties Estimated by In-Situ Data, Ground Based Remote Sensing and Modeling , https://www.mdpi.com/2072-4292/13/16/3163

**Detailed review:**

Introduction:

1. The authors motivate the readers about the current study in a way that MAX-DOAS measurements of near-surface concentrations and profiles are used to evaluate a global model CHASER (lines78-98). A study with such motivation is more suited for GMD (model evaluation papers). At least in the introduction, I could not find motivation for understating the atmospheric chemistry of the region of interest.

**Response:** We thank the reviewer for the comments. The following changes are included in response to this comment.

L62 – 73: The observational sites examined for the present study have different atmospheric characteristics. Thailand is strongly affected by pollution because of rapid economic development and urbanization. Moreover, biomass burning in Southeast Asia is a significant source of $O_3$ precursors, contributing up to 30% of the total concentrations during the peak burning season (Amnuaylorajen et al., 2020; Khodmanee et al. 2021). Because of rapid industrialization, India the second most populous country in the world, is witnessing an increasing $O_3$ trend along with $NO_2$ and HCHO concentrations in all major cities (Mahajan et al; 2015; Lu et al, 2018;). The Indo-Gangetic Plain (IGP), which covers ~21% of the Indian subcontinent land area is hotspots of severe air pollution (Giles et al; 2005, Biswas et al; 2019). In contrast, surface $O_3$ concentrations have shown an increasing trend in Japan, despite decreasing $NO_x$ and VOC concentrations related to emission control measures after 2000 (Irie et al., 2021). Therefore, observational and modeling studies must be conducted to improve our quantitative understanding of the $O_3$-$NO_x$-VOC relation in these regions.

2. Line 62 – I think it is more accurate to replace "radiation" with "radiance".

**Response:** Appropriate corrections has been included in the revised manuscript.

3. Lines 68-78: In my opinion, MAX-DOAS is, a powerful independent technique for monitoring atmospheric constituents, and I would mention it first before stating that it is complementary to in situ and satellite measurements. Observation, dataset and methods1.    In my opinion, the climate classification for Pantnagar should be done in a different way. The current classification does not consider summer as a separate season and is rather partly combined in spring and summer monsoon. The months Apr-June are extreme summer months in the Indo-Gangetic plain, with daytime temperatures above 40°C and an average temperature above 30 °C.

**Response:** Firstly, we have made appropriate changes to texts. Secondly, we thank the reviewer for the comment. The climate classification for the IGP region is based on many earlier studies in the region (i.e., Hoque et al, 2018b, Mahajan et al (2015) etc.). Such classification was adopted for consistency with the literature studies. Including summer as a separate season will not change the conclusions substantially.

**Observation, dataset and methods:**

4. Figure1: As the study focuses on the evaluation of the model over the south and east Asian region, I would recommend restricting the map boundaries to only relevant regions. The color codes show the surface volume mixing ratios (VMR) and not concentrations. The color bar legend should be corrected accordingly.

**Response:** Because, CHASER is a global model and global evaluation of the simulations has been included in the revised manuscript, we prefer not to change Figure 1. However, the color bar legends have been changed to "volume mixing ratio".

lines 139-141: Campaign is used two times in the same sentence.

**Response:** Appropriate corrections has been included in the revised manuscript.

5. MAX-DOAS system: What is the spectral range of the spectrometer used in these measurements. I am keen to know why the higher wavelength window of 460-490nm was chosen for $NO_2$ retrieval. The instrument used for this study participated in the CINDI and CINDI-2 campaign, and there the fit interval used for $NO_2$ retrieval was 425-490nm or 411-445nm.

**Response:** The spectral range of the spectrometer is 310 – 515 nm . While the CINDI (1 and 2) semi-blind intercomparisons used the window 425–490 nm, the present study uses 460- 490 nm for much faster retrievals by the DOAS fitting used in JM2. Also, the difference between representative wavelengths for $NO_2$ and $O_4$ can be very small, minimizing the wavelength-dependence of AMF information.

6. lines 148-149 Why would you want to minimize the variations in measured signals for various off-axis measurements. According to the DOAS principle, reference measurements should be taken at a 90° elevation angle to account for stratospheric contribution in the dSCDs. If the 90° measurements could not be taken due to any physical restrictions, this should be stated accordingly.

**Response:** Firstly, we used a spectrometer with a fixed integration time throughout the day. The intensity of the spectra usually depends on the elevation angle (EL) within a 15-min interval (time for a complete scan for all EL). Thus, the variation range of intensities measured at all ELs can be large, which occasionally leads to intensity saturation at the reference angle. To avoid such phenomenon, the refence measurement were conducted at 70˚ instead of 90˚. All the ELs were considered in the differential air mass factor

calculation to retrieve the vertical profiles. Thus, the choice of reference EL (70˚ or 90˚) is not a critical issue. Secondly, "variation in the measured signals" has been replaced with "to avoid saturation of intensity". The changes/addition in the revised manuscript is as follows:

L163-167: The sequences of the ELs at all the sites were repeated every 15 min. The reference spectra are recorded at EL of 70° instead of 90° to avoid saturation of intensity. Because all the ELs were considered in the box air mass factor ($A_{box}$) calculation to retrieve the vertical profile, the choice of reference EL (70˚ or 90˚) is not an important issue for this study.

7. How would the additional off-axis measurements at elevation angle > 10°reduce the systematic errors in the fitting results. In my opinion, measurements at some elevation angles (e.g. 15° and 30°), provide important information regarding the trace gas and aerosol profiles during inversion and should not be skipped if possible. Moreover, later in this study, the authors analyse profiles at high altitudes (> 2Km), and measurements at high elevation angles are necessary for the accuracy of such retrieval. Even the surface layer used in this study has a thickness of 1km, and measurements at high elevation angles are crucial for this layer as well.

**Response:** Restricting ELs < 10, has been adopted after the detailed study of Irie et al., 2015, utilizing MAX-DOAS, Sky radiometer, LIDAR and CRDS observations, in a view on requirement of a correction factors for the $O_4$ absorptions. Adopting a single correction factor ($f_{O4}$) of 1.25 for all of the elevation angles led to systematic overestimation of near-surface aerosol extinction coefficients. When the ELs were limited to ≤10° and an EL- dependent correction factor was used, the agreement between MAX-DOAS AEC profiles and other instruments were improved. With these modifications, the possible effects of temperature-dependent $O_4$ absorption cross section and uncertainty in DOAS fit on an aerosol profile retrieval are expected to minimize.

We also agree with the reviewer's opinion on the importance of higher ELs. Our current measurement and retrieval settings enhances the capability for observing the planetary boundary layer (PBL) as a result of the loss of sensitivity to extinction at high altitudes, where clouds are usually more dominant than aerosols. Thus, with our current setting realistic profiles can be derived, with lower accuracy in the higher altitudes.

Lines 170-171 and Figure 2: How does the DOAS fit for O4 look like in the two wavelengths window used in this study. An intercomparison of O4 dSCDs retrieval from the two fit windows should also be shown (at least in the appendix).

**Response:** The DOAS fit in the two-wavelength window is provided in the Supplementary information (Figure S1).

8. Line 187. It was difficult for me to visualise what the profile shapes look like for different values of F. It would be nice to have example plots showing the profile shapes for some values of F (similar to that shown in Fig 1 of Beirle et al., 2019 for h and s)

**Response:** Aerosol extinction profile with different F values has been shown in the supplementary information (Fig.S2)

9. It will be more accurate to save that VMRs are "calculated" using the partial VCDs rather than "converted". Though in this study, the height of the box is fixed, in general, it would be better to also mention that this conversion also considers the height of the box.

**Response:** We have replaced "converted with calculated". We did not assume fixed height but an exponential profile.

Why the heights of the boxes are chosen to be so wide at 1km. Several studies (e.g. Kumar et al 2020) indicate a strong gradient in NO2 profiles in the lowest 1km. As the MAX-DOAS measurements in this study are used to evaluation of near surface VMRs of trace gases from the global model, higher vertical resolution in the profile retrieval should be more relevant.

**Response:** We did not assume fixed height but an exponential profile. Considering this point and the fact that vertical resolution depends on the amount of aerosol loading, our retrieval settings are reasonable. Surface aerosol and $NO_2$ concentrations retrieved with such settings has shown good agreement with other observations and regional model (Damiani et al 2021, Irie et al, 2008). Our HCHO retrievals has been used to validate the TROPOMI observations (De Smedt et al 2021). Therefore, we think our retrieval settings for the model evaluation is reasonable.

10. Lines 222-223: In lines 170-171, the authors state that significant O4 absorption in 460-490nm was used to retrieve the O4 ΔSCD. Then why an aerosol retrieval in the same wavelength window is not performed? Rather an Angstrom exponent was used to retrieve the AOD at 470nm.

**Response:** We thank the reviewer for the comments. Aerosol at 476 nm are retrieved from the 460 – 490 nm. AOD at 470 nm is calculated using the Angstrom exponent for the $NO_2$ retrieval

11. Lines 223-223: What is the basis of the assumption of Angstrom exponent = 1. How does the choice of Angstrom exponent affect the retrieval?

**Response:** Angstrom exponent of 1 is assumed to reflect the strong wavelength - dependence of the AOD values. The choice of the Angstrom values had non- substantial impact on the retrieval. Uncertainty related to the Angstrom value was smaller than the uncertainties in the $A_{box}$ profiles. The following texts are added in the revised manuscript.

L 254 – 257: The choice of the Angstrom exponent value can induce uncertainty in the retrieved VCDs. However, such uncertainty was found to be non-significant compared to that of $A_{box}$ profiles. Uncertainty in the $A_{box}$ profiles are assumed to as high as 30 to 50%. Such values are derived empirically from comparison with sky radiometer and LIDAR observations (i.e., Irie et al., 2008b).

12. Line 261: Please cite the latest version of CHASER and mention the model version number.

**Response:** The latest version of the model is CHASER 4.0. Appropriated changes in the has been included in the manuscript.

13. Lines 262-264: What is the name of the chemical mechanism used for CHASER simulation?

**Response:** The chemical mechanism of CHASER is mainly adopted from the Master Chemical Mechanism (MCM). The information has been added in the revised manuscript.

L304 – 305 :The chemical mechanism is largely based on the master chemical mechanism (MCM, http://mcm.york.ac.uk)(Jenkin et al., 2015).

14. Please provide specific details of biomass burning emissions. Which product of ECMEF (might be GFAS?)

**Response:**

L321- 322: CHASER simulations. Anthropogenic emissions were based on the HTAP_v2.2 for 2008. Biomass burning and soils emissions from the ECMWF/MAC (Global Fire Assimilation System (GFAS)) reanalysis were used

15. Lines 278-285: Please provide an estimate of NOx and VOCs emissions from different sectors in the regions of interest. This is important to understand and confirm the important emissions sectors speculated in the subsequent sessions.

**Response:** The following texts were included in response to this comment

L332-342: NO$_x$ emissions in India were estimated as 14 Tg/yr in 2016, almost two-fold increase since 2005 (~8 Tg/yr), with the energy and transportation sector being the largest contributor (Sadavarte et al 2014). Indian anthropogenic non-methane VOCs (NMVOCs) emissions in 2010 were estimated ~ 10 Tg/yr , with respective contributions of 60, 16, and 12% from residential, solvents, and the transport sector( Sharma et al 2015). In Japan, vehicular exhausts (14 - 25%), gasoline vapor (9 - 16%), liquefied natural gas (7 - 10%), and liquefied petroleum gas (49 - 71%) contribute to the total VOC concentrations (Morino et al., 2011), with annual NMVOC emission of ~2 Tg (Kannari et al., 2007). Annual NO$_x$ emissions in Japan and Thailand in 2000 was estimated as ~2000 and 591 kt/yr, with the largest contribution from transport-oil use, followed by the energy and industrial sector (Ohara et al., 2007). Annual anthropogenic VOC emissions in Thailand are approximately 0.9 Tg, with 43, 38, and 20% contributed, respectively, from industrial, residential and transportation sectors (Woo et al; 2020).

16. Line 286: It would be nice to already mention here, what is the purpose of multiple CHASER simulations?

**Response:** The following texts has been included in response to this comment.

L343- 344: Multiple CHASER simulations with different settings used for sensitivity studies are presented in Table 3.

**Results and discussion:**

18. Lines 301 and 306 (and also at several sections of the manuscript): Figure 4 shows volume mixing ratios (not concentrations).

**Response:** Appropriate corrections has been included in he revised manuscript.

19. Figure 4: Please use the same y-axis scale for all the subplots. Also, in lines 302 and 348, it is important to mention that the standard deviations (or error bars) show the variability (not to be confused with measurement uncertainty).

**Response:** The y-axis scale of the HCHO plots are revised. A similar y-scale for the NO$_2$ plots cannot be used because the Chiba NO$_2$ levels are too high compared to Phimai and Pantnagar. Thus, only the NO$_2$ y-scale for Phimai and Pantnagar has been revised. Appropriate corrections have been included regarding the standard deviations.

20. Lines 326-328: How do these mixing ratios and the seasonality compare with the other studies reported in the Indo-Gangetic plain or other sites in India) (e.g. Biswas and Mahajan 2021, AAQR, Kumar et al 2020, ACP).

**Response:** The following texts were added

L391-399: The peak HCHO mixing ratio at Pantnagar is almost twice that of in Pune city (~ 3 ppbv) (Biswas and Mahajan, 2021), a site in the IGP region. The HCHO seasonality at the two sites are found to be dissimilar, because of differences in the VOC sources, however, lower mixing ratios during the monsoon is consistent. From another site in the IGP region (i.e., Mohali), Kumar et al., (2020) reported lowest HCHO VCDs during March 2014 and 2015, attributing them to lower biogenic and anthropogenic VOC emissions. At Pantnagar, the lowest HCHO mixing ratios are observed during the monsoon. The rainfall events in the IGP region shows strong annual variability (Fukushima et al. 2019). Discrepancies between the sites might be related to the rainfall pattern.

21. Section 3.1.3.1: In my understanding, the HCHO and NO2 indicator ratios ($R_{FN}$) indicator proposed by Martin et al., 2004 and Duncan et al., 2010 are based on the tropospheric vertical column densities (VCDs) and NOT concentrations. As the authors work with the MAX-DOAS system in this study, why they have chosen to calculate the ratio based on concentration and not the VCDs?

**Response:** Our MAXDOAS system is optimized for retrieving aerosol and trace gas information in the planetary boundary layer rather than across the entire tropospheric column. Therefore, the surface HCHO and $NO_2$ concentrations are used to calculate the ratios. Secondly, the effectiveness of the column-based $R_{FN}$ values are under discussion because of altitude-dependence of HCHO and $NO_2$. Thirdly, Souri et al. (2020)'s approach is based on surface $R_{FN}$ values, which has been adopted in this study.

22. Lines 390-391 and Figure 5: What is the person correlation coefficient of the scatter plots shown here. I wonder, how robust are the calculations drawn based on slopes of the scatter plot if the correlation is poor.

**Response:** The Pearson correlation coefficient is not critical in such case. Because, the slope indicates the transition region from VOC-limited to NOx limited, and is dependent on the $NO_2$-HCHO chemical feedback.

23. Figure 5: Please show a similar plot color-coded according to solar radiation (radiance at a selected wavelength). This would enable the authors to evaluate the contribution of chemistry in ozone production independent of available solar radiation.

**Response**: We thank the reviewer for the comments. Our discussion was primarily focused on the calculation of the transition line between the $NO_x$ and VOC-limited regions. Discussion on in-depth chemistry of ozone would be beyond the scope of our discussion. Thus, we did not include the suggested figure and related discussion in the manuscript. We will include such figures and discussion in a more detailed study.

24. How do the $R_{FN}$ values compare to previous studies (based on model, satellite and MAX-DOAS observations) in India (or Indo-Gangetic plain)?

**Response:** The following texts has been included in response to the comment

L499 – 510: At Pantnagar, high $O_3$ occurrences lie below (42%) and above (57%) the transition line, indicating that $O_3$ production is sensitive to both HCHO and $NO_2$ which contradicts results reported by Biswas et al. (2019). Based on satellite and ground-based observations, the study estimated the $R_{FN}$ values at a site in the IGP as > 4 and >2 respectively, and regarded the $O_3$ regime as $NO_x$-limited. Mahajan et al (2015) reported $R_{FN}$ values of less than 1 over the IGP region signifying as a VOC-limited region. Pantnagar is a sub-urban site situated beside a busy road. Therefore, effects of anthropogenic emissions are expected year-round, especially with pyrogenic emissions during the spring and post-monsoon period. $O_3$ sensitivity to both $NO_x$ and VOCs in the north-west IGP region has also been reported by Kumar and Sinha (2021). Therefore, the balance between the VOC and $NO_x$-limited region in the IGP is reasonable. The mean and minimum $R_{FN}$ value along the transition line are, respectively, 5.59 and 6.09. The minimum value (i.e., 5.59) is higher than Phimai (3.26), suggesting higher VOC levels at Pantnagar, consistent with the observations.

25. Line 472: I was wondering if the boundary layer height directly from the model simulations or reanalysis data products (e.g. ERA5) can be used and more suitable.

**Response:** The PBL heights from CHASER and reanalysis could be used, however, we have not tested the suitability of the datasets. For the current study, we followed the methodology suggested by Jin et al., (2017).

26. Section 3.1.3.2: It is difficult for me to understand the need to calculate the factor "F" (column to surface conversion factor, equation 9) in the context of this study. Authors use and discuss "F" to get column integrated values (i.e. concentration). However, the MAX-DOAS retrieval also provided the vertical column densities, which is a much simpler approach.

**Response:** As mentioned earlier, our system is optimized for retrieving aerosol and trace gas information in the planetary boundary layer rather than across the entire tropospheric column. Thus, the retrieved VCDs will mostly reflect the near surface information. Thus, the model values were used to calculate the F value.

27. Lines 501ß503 It might be true that there is no relevant literature in the south and south-east Asia presenting "F" values. But there is sufficient literature discussing both the surface concentrations and the vertical column densities, from which "F" can be derived.

**Response**: We agree with the reviewer's opinion and removed the sentence.

28. Lines 511-512: Averaging kernels are highly sensitive to atmospheric conditions, and hence these should be applied to individual profiles, and the averaging should be performed rather than using an averaged averaging kernel for a season.

**Response:** We have tested both ways – applying (1) AKs of individual retrieval, and (2) seasonal averaged AKs to the model profiles and found the results to be similar. Moreover, Franco et al (2015) applied seasonally averaged AKs to the model output. We have included the following text corresponding to this comment.

Applying individual AKs to the model outputs yielded similar results. The seasonally averaged AKs for both sites are shown in Fig S4.

29. Lines 515-516: How and between which parameters are the R values calculated? Are the R-Values calculated using individual measurements, daily average or seasonal mean?

**Response:** The R-value indicates the correlation between the observed and simulated HCHO seasonality. The text has been revised.

30. Figure 8: can the authors explain why the application of averaging kernels significantly decreases the column in Phimai, but results in an increase in Chiba?

**Response:** CHASER estimates at both sites are different, i.e., higher and lower compared to observations in Phimai and Chiba, respectively. Thus, the smoothing yields different results. Moreover, such differences can also potentially be related to the apriori values used for smoothing. The apriori values are obtained from the retrieved SCD and VCD values. The differences in the atmospheric conditions at both sites will be reflected in the apriori values, and thus leading to different smoothing results.

31. Lines 533, 550, 590, 652, 653, 655: The MAX-DOAS profile retrievals are performed at a vertical resolution of 1Km, and hence it is not appropriate to quantitively evaluate the model profiles at intermediate layers (e.g. 0.5 km or 200m).

**Response:** The evaluation at the intermediate layers has been removed from the discussion.

32. Line 546: Please provide appropriate reference justifying the model overestimation of biogenic emissions.

**Response:** The texts has been edited in response to this comment

L771-773: Consequently, it is likely that the biogenic emissions for this region in the model are overestimated. The Southeast Asian isoprene emissions in CHASER is 128 Tgyr$^{-1}$, higher than the CMAS-GLO-BIO (Sindelarova et al., 2022) inventory (78 Tgyr-1).

Lines 554-558: If biogenic emissions are overestimated in the model (as mentioned before), I would expect a higher increase in simulated HCHO than observed between January and August.

**Response:** We have edited the earlier sentence. The biogenic emissions for the Southeast Asian region are overestimated in CHASER. Because the observed HCHO columns are enhanced during the dry season, the difference between the observations and simulations are low.

33. I am surprised to learn that emissions due to wintertime heating is not included in the anthropogenic emission inventory. From the EDGAR website (https://edgar.jrc.ec.europa.eu/dataset_htap_v2#p1), it seems that the sectors "htap_6 Residential" and "htap_3 Energy" include the wintertime heating emissions.

**Response:** We thank the reviewer for pin pointing the issue. We have revised the sentences.

34. Sector 3.2.3 could be merged with 3.2.1 and 3.2.2.

**Response:** We prefer to keep the subsections for better readability

35. Lines 678-680: It is not clear for me, why observations above 1.8km are compared with the model. Both MAX-DOAS profile retrievals and model simulation are performed above the ground level.

**Response**: The comparison between the model and measurements at the Pantnagar site has been discarded.

36. Lines 685-686: This brings me to the previous comment. Why in the first place, measurements are restricted to elevation angles less than 10°.

**Response:** The discussion related to this comment has been discarded. The use of ELs < 10 has been explained in the earlier comments

37. Lines 687-691: What is included in the whole IGP. Please show it on a map. What are the limitations of comparing the measurements at a point (representative of a few Km) to the entire IGP?.

**Response:** Instead of a map, we have added the following texts:

L905-913: The IGP is the most fertile region in South Asia, which accounts for approximately 50% of the total agricultural production of India and is one of the significant contributing regions to the global greening based on leaf area index (Sarmah et al., 2021). Moreover, IGP is one of the regional HCHO hotspots in India (Chutia et al., 2019). The observed HCHO seasonality at Pantnagar is consistent with that reported by Mahajan et al. (2015) for the entire IGP region. Consequently, comparison with the HCHO retrievals in Pantnagar can assess the model capability in the IGP region. The spatial representativeness is a limitation for comparison between a point measurement and regional simulations. Thus, the results are interpreted qualitatively.

38. Figure 11: Please use the same y-axis range for subplots of HCHO and NO2.

**Response:** The discussion related to comparison in Pantnagar has been discarded in the revised manuscript.

39. Line 711-712: If the biogenic emissions are overestimated, how come the simulated isoprene concentrations are reasonable?

**Response:** We have revised the earlier sentences that mentioned the "biogenic emission in the model is overestimated" to "biogenic emissions for Southeast Asia in the model is overestimated". The Indian isoprene emissions in CHASER is 15 Tg/yr, comparable to the MEGAN estimate of 12 Tg/yr. (Guenther et al; 2006). CHASER simulated isoprene concentrations in the IGP region are also comparable with the observations. Thus, the simulated isoprene concentrations in the IGP region is deemed reasonable.

40. Figure 12 and line 735: In my opinion, it will be better to show the time series at the three stations rather than the zonal mean if the inferences are made with respect to observation at the three sites.

**Response:** We have revised figure 12, which is fig.14 in the revised manuscript and the related discussions following the comment in section 3.3.4.

41. Lines 738:740: 10% is the average, and based on this, one cannot infer that the comparison result will improve by at least 10%.

**Response**: We thank the reviewer for pinpointing the issue. We have removed the sentence.

42. Line 748: How did the authors estimate that the impact of model resolution is 20%.

**Response:** The impact was estimated based on the differences between the two simulations. However, the sentence has been removed in the revised manuscript.

43. Line 755: What stops the authors from using an updated emission inventory if those are already available.

**Response**: We thank the reviewer or the comments. The information has been taken from a study in the literature (Miyazaki et al., (2020)). Because of some technical issues, we haven't yet updated the emission inventories, which we are currently trying to fix. Uncertainty related to the old emission inventory are discussed in the light of comparison with the OMI data in section 3.3.4

44. Line 782: How are the biogenic emissions optimized?

**Response:** We used multiple simulations with varying biogenic emissions. The biogenic emissions were changed until the wet season estimates coincides with the observations.

45. Lines 789-791: Please provide an estimate of NOx emissions from different sectors based on the emission inventory used for the simulations.

**Response:** We have added the information in section 2.3. The following texts are included:

L328 – 331: Global $NO_x$ emissions of 43.80 $TgNyr^{-1}$ are used in the simulations, considering industries (23.10 $TgNyr^{-1}$), biomass burning (9.65 $TgNyr^{-1}$), soil (5.50 $TgNyr^{-1}$), lightning (5 $TgNyr^{-1}$), and aircrafts (0.55 $TgNyr^{-1}$) as significant sources. Global isoprene emissions from vegetation were set to 400 $TgCyr^{-1}$.

---

## Author Response (AR2)

**Response to reviewer and editorial comments**

**Response to reviewer comments**

We thank the referee for the helpful comments. We have revised the manuscript accordingly. The responses (blue fonts) are provided after stating the reviewer comments. Figure, Table, and line numbers correspond to the revised manuscript. The highlighted text are corresponding changes in the revised manuscript.

**Major comments:**

The authors stated in their reply to Reviewer #1 that they removed Pantnagar from the model evaluation. However, looking at Sections 3.3.3 and 3.3.4, it seems not to be the case. A clarification is needed here.

**Response:** We thank the reviewer for the comment. For the Pantnagar site, the comparison (profiles and partial columns) between the observations and the simulations close to the measurement site have been removed. In section 3.3.3, HCHO observation are compared qualitatively with the simulations in the IGP region. In section 3.3.4, simulations at two different spatial resolutions for the site are compared. Overall, the point to point comparison for the Pantnagar site has been discarded.

A discussion on the impact of using more recent inventories in the CHASER simulations on the comparison results with OMI is missing and should be added (see also related specific comment below).

**Response:** We thank the reviewer for the comments. Yes, surely, including such comparison studies will add more merit to the discussions. However, as mentioned earlier, currently we are unable to implement updated inventories in our simulations due to some technical limitations, which we are working to fix. To address this aspect, we have included the following text in the revised manuscript.

L576- 579 It should be noted that simulations based on old $NO_x$ emission inventory will likely affect the model-satellite comparison results. However, the current study has not assessed such impact due to technical issues related to using an updated emission inventory. This issue will be addressed in a separate study.

**Specific comments:**

Figure 1, page 6: May be replace 'during June in 2018' by 'in June 2018' in the title?

**Response:** Appropriate corrections has been included in the revised manuscript.

Page 11, line 213: not clear what you did here. Did you fix your k value at 100km to the mean SAGE III extinction coefficient in the 15-40km altitude range? A clarification is needed here.

**Response:** Yes. According to Irie et al. (2008, 2011, 2015), such assumptions has negligible impact on the retrieval due to lower sensitivity of the MAX-DOAS observations above 2 km.

Page 11, line 213: may be replace 'non-substantial' by 'negligible'?

**Response**: Appropriate corrections has been included in the revised manuscript.

Figure 3, page 13: Do the error bars on the averaging kernels correspond to the standard deviation? This information should be included in the legend of the figure.

**Response:** The caption of figure 3 has been revised

Page 14, lines 283-284: not clear to me how the cloud screening approach works. More particularly, how can you retrieve information on clouds based on the HCHO and NO2 dSCD residuals? This point should be clarified.

**Response:** It is known that clouds can bias the retrieved concentrations. While the discrimination between clouds and aerosols is still very challenging, the following data screenings were made to minimize the influence of clouds. First, we filtered output from the retrieval only for retrieved AOD less than 3, the largest value in the LUTs. This excludes large optical depth cases, most of which should be due to optically thick clouds. Further data screening was made using the root-mean squares of residuals of the $O_4$, $NO_2$, and HCHO dSCDS. Larger residuals likely occur when constructing a profile is too simple to represent the true profile, particularly with a steep vertical gradient of extinction due to clouds. Also, rapid changes in optical depth within 30 min that corresponds to the full scanning time can lead to larger residuals. The threshold values were determined statistically corresponding to the mode plus one sigma ($1\sigma$) in the logarithmic histogram of relative residuals. The following sentences has been included in the revised manuscript.

L282-285: Larger residuals likely occur due to two reasons: (1) when the constructed profile is too simple to represent the true profile, particularly with a steep vertical gradient of

extinction due to clouds, and (2) rapid changes in optical depth within 30 min (time for one complete scan) (Irie et al, 2011).

Page 17, line 253: please remove 'For analysis,' at the beginning of the sentence.

**Response:** Appropriate corrections has been included in the revised manuscript.

Page 18, line 365: I would replace 'signifying' by 'indicating'.

**Response:** Appropriate corrections has been included in the revised manuscript.

Page 22, line 461: I would add the following sentence (or something similar): 'This criterion on the SZA is also applied for the selection of the NO2 and HCHO concentrations.

**Response:** Appropriate corrections has been included in the revised manuscript as follows:

L463: This criterion on the SZA is also applied for the selection of the $NO_2$ and HCHO concentrations.

Page 22, line 462: In order to avoid confusion, you should mention that the good agreement between the JM2 O3 product and ozonesondes was obtained in a previous study and has not been checked here. I would rephrase the sentence as 'Although not checked here, the JM2 O3 product showed good agreement with ozonesonde measurements in a previous study (Irie et al., 2021).'

**Response:** Appropriate corrections has been included in the revised manuscript.
L464-465: Although not checked here, the JM2 $O_3$ product showed good agreement with ozonesonde measurements in Tsukuba (Irie et al., 2021).

Figure 5 (c) and (f), page 23: in order to better distinguish the data points, you could use a y-axis upper limit of about 12 ppbv instead of 20 ppbv?

**Response:** Figure 5(c) and (f) have been revised.

Page 23, line 489: 'Schroder' -> 'Schroeder'

**Response:** Appropriate corrections has been included in the revised manuscript.

Page 27, lines 572-574: Given the fact that the comparison OMI versus CHASER is done at a global scale, it is not clear to me why only few days with OMI observations are remaining in July and December after filtering.

**Response:** We thank the reviewer for the comments. We selected the coincident dates between the simulation and daily observations for every month based on fixed data filtering criteria. Unfortunately, the NO$_2$ data selection results yielded very few coincident days in July and December, thus discarded from the comparison. The word" coincident" has been added to avoid confusion as follows:

L574-575 The month of July and December were discarded from the NO$_2$ comparison because few ==coincident== days (only five days) were available after filtering.

Page 27, introductory paragraph on the comparison with OMI (lines 568-574): at the end of this paragraph, I would add a disclaimer about the fact that the comparison results are likely affected by the use of rather old emission inventories in the model simulations. I would then add a Section 3.2.3 with a discussion on the impact of using more recent inventories on the OMI versus CHASER comparison results (a bit like the authors did in Section 3.3.4 for the comparisons at the MAX-DOAS sites).

**Response:** We thank the reviewer for the comments. Yes, surely, including such comparison studies will add more merit to the discussions. However, as mentioned earlier, currently we are unable to implement updated inventories in our simulations due to some technical limitations, which we are working to fix. To address this aspect, we have included now the following text in the revised manuscript.

==L576- 579 It should be noted that simulations based on old NO$_x$ emission inventory will likely affect the model-satellite comparison results. However, the current study has not assessed such impact due to technical issues related to using an updated emission inventory. This issue will be addressed in a separate study.==

Page 27, line 582: 'The spatial representativeness between…' -> 'The difference in spatial representativeness between…'; 'observation' -> 'observations'; 'one potential reasons' -> 'one potential reason'

**Response:** Appropriate corrections has been included in the revised manuscript.

Page 27, line 583: I don't understand why the word 'however' is used here. The fact that the CHASER simulations at 1.1° improve the MBE and RMSE is a further indication that the difference in spatial representativeness between the model and observations is one potential reason for the observed negative bias.

**Response:** The word "however" has been removed.

Page 28, line 592: the second 'although' in the sentence should be removed.

**Response:** Appropriate corrections has been included in the revised manuscript.

Figure 7, page 29: Why the CHASER NO2 and HCHO maps are not shown in the figure?

**Response:** Model simulations have been included in Figure 7.

Page 32, line 656: It should be 'Figure 9' instead of 'Figure 7'.

**Response:** Appropriate corrections has been included in the revised manuscript.

Page 38, lines 795-796: I would start the sentence as follows (or something similar): 'Overall, given the large uncertainty on the MAX-DOAS profiles (see Fig. 10), the differences ….'

**Response:** The sentence has been revised.

Page 41, line 848: Referring to Fig. S5 is not correct (it corresponds to the discussion on the correlation between HCHO concentration in the 0-2km altitude range and temperature). So the figure on the impact of the MAX-DOAS a priori profile on the smoothing of the CHASER NO2 profile at Chiba seems to be missing. When you will add this figure, please correct the figures numbering in the Supplement and in the main text accordingly.

**Response:** We thank the reviewer for the comment. We have added the figure (Fig S7) in the supplement and respective figure numbers in the manuscript have been revised.

Page 45, line 910: 'observatios' -> 'observations'
**Response:** Appropriate corrections has been included in the revised manuscript.

Page 54, line 1094: I would give the literature reference (Duncan et al., 2010) associated to the standard transition region approach.
**Response:** The reference has been included in the conclusion.

Page 54, line 1099: 'clarified' -> 'further indicate'
**Response:** Appropriate correction has been included in the revised manuscript.

Page 54, line 1102: I would replace 'agreed well' by 'agreed reasonably well'.

**Response:** Appropriate correction has been included in the revised manuscript.

Page 56, Acknowledgements: Personally, I would also thank the OMI HCHO and NO2 data providers.

**Response:** The data providers have been acknowledged
* * *
**Response to Editorial Comments**

\* please check the suggestions made by the reviewer carefully, and implement them

**Response:** We have addressed all the reviewer comments within our knowledge and capability.

\* please check the last abstract of your summary as it needs some language editing

**Response:** The abstract has been revised.

\* in Figure 2, the label of the y-axis says "differential OD" while the quantity shown appears to be absolute OD

**Response:** We think the y-axis caption is correct. The spectra are plotted as the differential optical density from the reference spectrum. Similar type of figure has been used in our previous works also (i.e., Irie et al., 2011, Hoque et al, 2018)

\* in the caption of Figure 3, please indicate what the error bars represent

**Response:** The caption of Figure 3 has been revised.

\* in Figure 4, a bit more vertical space between the sub-figures would improve readability

**Response:** Figure 4 has been revised

\* in Figure 7, the use of colour schemes appears arbitrary. I would suggest to use the colour scheme from (b) for both difference plots and to make sure that it is centred on 0 making yellow the value shown for 0

**Response:** Figure 7 has been revised

* in Figures 8, 9, 14, 15 and 16, I think it would be good to always include the 0 in the y-axis to facilitate comparisons

**Response:** Figure 8,9,14,15, and 16 have been revised.